# River patterns reveal two stages of landscape evolution at an oblique convergent margin, Marlborough Fault System, New Zealand

Alison R. Duvall[1], Sarah A. Harbert[1*], Phaedra Upton[2], Gregory E. Tucker[3], Rebecca M. Flowers[3], and Camille Collett[1**]

[1]Department of Earth and Space Sciences, University of Washington, Seattle, 98195, USA
[2]GNS Science, Lower Hutt, 5040, New Zealand
[3]Department of Geological Sciences, University of Colorado, Boulder, 90309, USA
*now with Pacific Lutheran University, Parkland, 98447, USA
**now with the United States Geological Survey, Golden, 80401, USA

*Correspondence to*: Alison R. Duvall (aduvall@uw.edu)

**Abstract.** Here we examine the landscape of New Zealand's Marlborough Fault System, where the Australian and Pacific plates obliquely collide, in order to study landscape evolution and the controls on fluvial patterns at a long-lived plate boundary. We present maps of drainage anomalies and channel steepness, as well as an analysis of the plan-view orientations of rivers and faults, and find abundant evidence of structurally-controlled drainage that we relate to a history of drainage capture and rearrangement in response to mountain building and strike-slip faulting. Despite clear evidence of recent rearrangement of the western MFS drainage network, rivers in this region still flow parallel to older faults, rather than along orthogonal traces of younger, active strike-slip faults. Such drainage patterns emphasize the importance of river entrenchment, showing that once rivers establish themselves along a structural grain, their capture or avulsion becomes difficult, even when exposed to new weakening and tectonic strain. Continued flow along older faults may also indicate that the younger faults have not yet generated a fault damage zone with the material weakening needed to focus erosion and reorient rivers. Channel steepness is highest in the eastern MFS, in a zone centered on the Kaikōura Ranges, including within the low-elevation valleys of main stem rivers and at tributaries near the coast. This pattern is consistent with an increase in rock uplift rate toward a subduction front that is locked on its southern end. Based on these results and a wealth of previous geologic studies, we propose two broad stages of landscape evolution over the last 25 million years of orogenesis. In the eastern MFS, Miocene folding above blind thrust faults generated prominent mountain peaks and formed major transverse rivers early in the plate collision history. A transition to Pliocene dextral strike-slip faulting and widespread uplift led to cycles of river channel offset, deflection and capture of tributaries draining across active faults, and headward erosion and captures by major transverse rivers within the western MFS. We predict a similar landscape will evolve south of the Hope fault, as the locus of plate boundary deformation migrates southward into this region with time.

## 1 Introduction

Tectonics and deformation impart a lasting impression on landscapes and often act as primary drivers of surface processes (e.g. Wobus et al., 2006; Whittaker et al., 2008; Cowie et al., 2008, Kirby and Whipple, 2012; Whipple et al., 2013). Rivers, in particular, are influenced strongly by tectonic forces, as they respond to mountain uplift (e.g. Whipple and Tucker, 1999; Bishop, 2007), lateral advection of crust (e.g. Miller and Slingerland, 2006; Duvall and Tucker, 2015; Gray et al., 2017) and to material weakening along faults (e.g. Koons, 1995; Koons et al., 2012; Roy et al., 2015; 2016a). Since the early days of river profile analysis (Hack, 1957; 1973; Seeber and Gornitz, 1983, Snow and Slingerland, 1987; Whipple and Tucker, 1999), measures of channel steepness, concavity, and knickpoints have served as the main geomorphic metrics to resolve uplift and fault-slip histories from the landscape in deforming terrain. Planform patterns of drainage networks, from offset channels (Wallace, 1968) to rivers flowing around or through folds (Keller et al., 1998), can also yield valuable insights into the tectonics underlying the landscape response. The planform rotation of river basins has been used as a marker of crustal strain (Hallet and Molnar, 2001; Ramsey et al., 2007; Castelltort et al., 2012; Guerit et al. 2016) and to assess the style and rates of off-fault deformation (Goren et al., 2015; Gray et al., 2017). Drainage network morphometry may yield additional evidence of reorganization of catchments by ridge migration (Pelletier, 2004; Willet et al., 2014) or by wholesale river capture (Craw et al., 2003; Clark et al., 2004; Craw et al., 2013) and flow reversal (Benowitz et al., 2019), often in response to tectonics. Drainage anomalies, or unusual patterns in river planform, can serve as recorders of these past river captures and drainage reorganizations (e.g. Brookfield, 1998; Hallet and Molnar, 2001; Burrato et al., 2003; Clark et al., 2004; Delcaillau et al., 2006; Willett et al., 2014), though not all anomalies necessarily link to river captures, even in active tectonic settings (Bishop, 1995).

Recent studies focusing on lithology and fractures in bedrock channels have also revealed the importance of material strength in setting incision rates and response times of channels to base level forcings (e.g. Stock and Montgomery, 1999; Whipple and Tucker, 2002; Cook et al., 2009; Whittaker and Boulton, 2012; Forte et al., 2016; Yanites et al., 2017; Dibiase et al., 2018; Zondervan et al., 2020) as well as in setting patterns in the position and orientations of rivers (e.g. Koons et al., 2012; Roy et al., 2015; 2016a; 2016b; 2016c; Upton et al., 2018; Scott and Wohl, 2019). Detailed field and numerical modeling studies of rivers in faulted landscapes in particular demonstrate the sensitivity of fluvial incision to gradients in erodibility between weak fault zones and the surrounding stronger bedrock (Roy et al., 2016c). These studies further show that structurally aligned drainages, with anomalously straight reaches, originate in response to the localization of fluvial erosion along the zones weakened by tectonic strain (Roy et al., 2015; 2016a;2016b).

Understanding the interaction and feedbacks among faulting, uplift, and erosion is especially important at plate boundaries that experience long-lived, complex, and changing deformation fields over orogenic timescales. Progress on these fronts has been made through studies of long-term river evolution in collisional orogens such as eastern Tibet (Clark et al., 2004; Wang et al., 2017) and the Andes (Horton et al., 2001; Garcia and Hérail, 2005; Struth et al., 2015), and in fold-thrust belts such as

the High-Atlas Mountains of Morocco (Babault et al., 2012), the Zagros Mountains of Iran (Ramsey et al., 2008), and the Otago region of New Zealand (Jackson et al., 1996; Amos et al., 2010), as well as in experimental drainage networks (Lague et al., 2003; Viaplana-Muzas et al., 2015; Guerit et al., 2016; Viaplana-Muzas et al., 2019). Collectively, these and other studies show that if we wish to extract tectonic information from landscapes at plate-boundary settings with long histories, we must understand how the drainage network evolves as the orogen develops. In doing so, we require specific knowledge of the timescales of river response to the changing tectonic boundary conditions and must assess if and when patterns in the drainage network are overprinted as mountains uplift, older faults change, and new faults form. We must also parse the importance of river capture processes and the development of topography along faults in the establishment of or destruction of structurally aligned drainages.

In this contribution, we examine river patterns across the NE South Island, New Zealand's Marlborough Fault System (MFS), at the southern end of the Hikurangi Subduction Zone (Fig. 1). The MFS landscape has evolved in the face of changing oblique Pacific-Australian relative plate motion over the past 25 million years (e.g. King, 2000) and thus offers an excellent case study of how landscapes record long-lived deformation and in particular, what sets the patterns in drainage networks in complicated and evolving tectonic settings. In this study, we present analysis of topography, fluvial morphologies in planform and profile, and orientations of rivers as compared to active and inactive faults. We combine these observations with existing geologic data to discuss the controls of fluvial patterns and to assess the long-term landscape evolution of the MFS.

## 2 Geologic Background: Marlborough Fault System

### 2.1 Geologic Setting

Steep, rugged mountains cover much of the field site, with major Marlborough faults cutting across this topography (Fig. 2, Fig. S1). Faults commonly slice the lower foothills of eastern range fronts, expressing steep, faceted hillslopes. The fault segments tend to lie a few hundred meters above the main rivers, rather than bounding mountain fronts proper (Fig. 1c). Fault slip and mountain uplift occur across the MFS in response to recent and ongoing deformation within New Zealand that results from the oblique convergence of the Australian and Pacific plates (DeMets et al., 2010). This complicated tectonic boundary includes westward oblique subduction of the Pacific plate beneath the Australian plate along the Tonga-Kermadec-Hikurangi trench off the east coast of the North Island and eastward subduction of the Australian plate beneath the Pacific plate along the Puysegur Trench off the southern coast of the South Island (Fig. 1a). In between, the dextral-reverse Alpine Fault and the ~150 km wide MFS together link these two subduction zones in a broad transform boundary (Walcott, 1978).

At present, relative plate motion within the MFS is thought to be taken up primarily on a suite of four parallel, mostly dextral faults, the Wairau, Awatere, Clarence and the Hope-Jordan-Kekerengu system, that slip at rates of ~5 to 25 mm/yr based on offset of Holocene and Quaternary features (Knuepfer, 1984, 1988; Cowan, 1989; Van Dissen & Yeats, 1991; McCalpin, 1996;

Little & Roberts, 1997; Little et al., 1998; Benson et al., 2001; Nicol & Van Dissen, 2002; Langridge et al., 2003; Mason et al., 2006; Langridge et al., 2010, Little et al., 2018) and geodesy (Wallace et al., 2007). In addition to these major faults, several subsidiary reverse and strike-slip faults stretch across the region, both onshore and off, including the 20+ fault strands that ruptured during the 2016 $M_w$ 7.8 Kaikōura Earthquake (Litchfield et al., 2018) (Fig. 1). The MFS, which splays northeastward from the northern end of the Alpine fault, straddles the transition from subduction to oblique continental plate collision. Subduction terminates offshore to the south and east of the Kaikōura peninsula, adjacent to the thinned continental crust of the southward migrating Chatham Rise (Fig. 1a). Geophysical imaging shows that Pacific plate lithosphere extends beneath the MFS (Eberhart-Phillips & Reyners, 1997; Eberhart-Phillips & Bannister, 2010). Prior to the 2016 earthquake, this part of the subduction zone was presumed to be strongly locked and inactive (Eberhart-Phillips & Bannister, 2010). Recent InSAR, seismology, and GPS data from this event reopened debate regarding earthquake activity on the subduction interface (e.g. Hamling et al., 2017; Bai et al., 2017; Wallace et al., 2018; Lanza et al., 2019).

## 2.2 Plate Tectonic History

Over the ~ 25 Ma lifespan of the Kaikōura Orogeny, the Pacific-Australian plate boundary has rotated clockwise and propagated southward (Walcott, 1998; King, 2000; Cande & Stock, 2004), creating a complex and evolving deformation field within the greater Marlborough region. Patterns in low-temperature thermochronology point to two general phases of Cenozoic exhumation related to the evolving Pacific-Australian plate boundary in this region. Early in the plate boundary history (Late Oligocene to Early Miocene), deformation appears to have been focused on a few important thrust faults and folds that generated topography in the Kaikōura Ranges (Baker and Seward, 1996; Collett et al., 2019). Spatial patterns in apatite and zircon (U-Th)/He ages and thermal modeling reveal Miocene cooling localized to hanging wall rocks, first along the Clarence fault in the Inland Kaikōura Range, then along the Jordan fault in the Seaward Kaikōura Range (Collett et al., 2019 – see sample locations in Fig. S1). A major influx of coarse Neogene clastic sediment within the Marlborough region (Rattenbury et al., 2006), including the Early Miocene Great Marlborough Conglomerate (Reay, 1993; Fig. S1), provides sedimentary evidence of exhumation in response to increased convergence along the plate boundary (Walcott, 1978) and to the emergence of this part of New Zealand above sea level (Browne, 1995). Prior to the development of the current Pacific-Australian plate boundary, much of the New Zealand continent was submerged beneath the ocean (Sutherland, 1999).

Since the late Miocene / early Pliocene, the main Marlborough faults display a primary right-lateral sense of slip with a small component of vertical displacement on the order of 10-15:1 horizontal slip to vertical slip, based on measurements of displaced landforms and slickenside striations on active fault traces (Nicol and Van Dissen, 2002). Low-temperature thermochronology shows a progression to widespread, rapid exhumation starting at ~5 Ma (Collett et al., 2019). Translation of crust along the Marlborough faults into a subduction front termination on its southern end (Little & Roberts, 1997; Walcott, 1998) is thought to have generated vertical block rotations and driven widespread rock uplift and new fault development during this phase of orogeny (Collett et al., 2019).

The southwestward migration of the Chatham Rise with respect to the Australian Plate is thought to have driven the southward propagation of dextral faulting within this time period (Little & Roberts, 1997; Furlong, 2007; Wallace et al., 2007; Furlong & Kamp, 2009). Estimates of timing of fault onset from total offset of dated features shows a north to south progression in dextral motion, with right-lateral faulting starting first on the Awatere fault at ~7 Ma (Little and Jones, 1998), followed by the Clarence fault at ~3 Ma (Browne, 1992), and on the Hope fault by ~1 Ma (Wood et al., 1994). The same faults show a spatial trend in slip rate, with the Hope fault located farthest to the south and slipping presently at the fastest rate (20 – 25 mm/yr). Incipient faults and mountain building are concentrated south of the Hope fault, with the next major Marlborough strike-slip fault hypothesized to coalesce through this region in the future (Cowan et al., 1996; Wallace et al., 2007). When this happens, slip rates on the Hope fault would slow down to the background slip rate of 2-5 mm/yr presently exhibited by the other Marlborough faults and the new fault would slip fastest as it would be the main link between the Alpine fault and the Hikurangi subduction zone.

## 2.3 Study Area Topography

The study area can be divided into three broad geomorphic domains based on general patterns in topography (Fig. 2). The eastern part of the study area contains the highest peaks (~2.5 km) and relief (Fig. 2, profiles 1 and 2). Here the parallel Inland and Seaward Kaikōura Ranges construct a dramatic boundary with the Pacific Ocean. These ranges, and a similar swath of mountains just north of the Awatere fault and valley, are separated by the Awatere and Clarence rivers, which flow eastward, parallel to the range fronts, through wide, linear valleys (Fig. 1,2). In the western part of the study area, elevations are more consistent across the landscape (~1.5 km) and relief is lower in the absence of the valleys and prominent peaks of the Kaikōura Mountains (Fig. 2, profiles 3 and 4). This region contains the headwaters of the Awatere and Clarence rivers. In its upper reaches, the ~400 km long Clarence river flows mainly north to south, flowing across the E-W trending Marlborough faults, before turning east to flow through the Kaikōura domain to the Pacific Ocean (Fig. 1). In contrast, the Awatere river flows eastward along its entire course. The third domain lies south of the Hope fault and includes the lower-elevation Hundalee Hills (max elevation ~ 500 m) and smaller drainage catchments orthogonal to the coast (Fig. 2,3).

## 3 Methods

In this study, we analyze trends in MFS rivers and faults in order to evaluate the fluvial response to Kaikōura Orogeny deformation and to assess the main drivers of drainage network patterns. We compare results of these analyses among the three geomorphic domains based on study area topography described above (section 2.3).

### 3.1 Planform Analysis of Rivers

We examined planform river patterns across the study site in order to find evidence of patterns in drainage indicative of fluvial disruption. We used 1:50000 topographic maps from Land Information New Zealand (LINZ), Google Earth imagery, and field

observations to identify anomalous river reaches, those that deviate from common dendritic drainage patterns (Zernitz, 1932), in the following categories: river elbows, locations where rivers take sharp (~ 90°) bends (McCalpin, 1996; Craw and Waters, 2007), obtuse tributaries, where the channels meet the main river at angles 90° or greater and barbed tributaries, which are a form of obtuse tributary where the 90° or greater channel junction occurs due to a deflection of the channel such that the

tributary flows into the larger river in an upstream rather than downstream direction (Hackney and Carling, 2011) and water gaps, where sections of river flow across mountain divides (Keller et al., 1998) (Fig. 3).

### 3.2 Fault and River Orientation Analysis

In order to assess the importance of fault development in dictating the flow path of the drainage network across the study site,

we quantified the planform orientation of faults and rivers using ArcGIS and TopoToolbox 2 (Schwanghart and Scherler, 2014). Faults were compiled from the GNS Science database (Langridge et al., 2016), which divides them into active and inactive categories. We adopt the GNS standard that considers a fault to be active if it shows evidence of movement at least once in the last 100,000 years. The river network was generated from 8m digital elevation data available from the New Zealand government (LINZ, 2012). Within the three study-area domains, we computed orientations of river segments of second and

higher Strahler order (Fig. 3b) and individual fault segments from the two activity categories (Fig. 4b). We avoided first-order river tributaries in this analysis to reduce noise in the dataset. We computed mean fault orientations using the Linear Directional Mean tool in ArcGIS. We computed river orientations from river network segment routines contributed to TopoToolbox by Philippe Steer (https://topotoolbox.wordpress.com/2016/05/13/orientation-of-stream-segments/), which identifies river segments based on locations of channel heads, confluences, and outlets. Mean orientations of both fault and river features were

computed based on start and end point coordinates of each feature segment. To avoid short feature segments having undue influence on statistics, prior to plotting and compiling the data, we weighted segment orientations by a unit length defined by the DEM grid cell size. We then used CircStat, a MATLAB toolbox for circular statistics (Berens, 2009), to plot output as polar histograms (Fig 3, 4) and to calculate statistics for fault and river orientations (Table 1). In these computations and plots, 0° is east-west, 90° is north-south, and 180° is west-east.

### 3.3 River Profile Analysis

Empirical studies of bedrock channels in a variety of settings typically show a scaling between local channel slope and upstream drainage area $S=k_s A^{-\theta}$ where S is the local channel gradient, A is the contributing drainage area, and $k_s$ and $\theta$ are the channel steepness and concavity indices, respectively (Flint, 1974; Wobus et al., 2006). Channel steepness may vary dramatically from channel to channel, reflecting differences in rock uplift rate and bedrock lithology most commonly (e.g.

Kirby and Whipple, 2001; Snyder et al., 2003; Kirby et al., 2003; Duvall et al., 2004; Harkins et al., 2007; Ouimet et al., 2009; DiBiase et al., 2010; Cyr et al., 2014), but also may vary along a single channel profile where an upstream convexity separates channel reaches with different steepness. Channel concavity, θ, on the other hand, is relatively constant among

well-adjusted, quasi-equilibrium channels (e.g. Kirby and Whipple, 2001; Tucker and Whipple, 2002; Snyder et al., 2003; Kirby et al., 2003; Duvall et al., 2004; Whipple and Meade, 2004; Wobus et al., 2006; Harkins et al., 2007) but can vary strongly in landscapes that have experienced a complex history of forcing by tectonic, climatic, or lithologic variation (Dorsey and Roering, 2006; Whittaker et al., 2008).

We used the Topographic Analysis Kit (TAK, Forte and Whipple, 2019), which works with TopoToolbox (Schwanghart and Kuhn, 2010; Schwanghart and Scherler, 2014), to generate a batch normalized channel steepness index ($k_{sn}$) map and χ map of the study area (Fig. 5). $K_{sn}$ is a routinely used measure of channel steepness that accounts for drainages of different profile shape by using a reference channel concavity (Wobus et al., 2006). The variable χ has been developed as a proxy for steady-state channel elevation (Willett et al., 2014) that comes from an integral transformation of the river profile's horizontal coordinates and has units of distance (Perron and Royden, 2013). χ, or more specifically, differences in χ across drainage divides, has been used as a measure of river basin disequilibrium (Willett et al., 2014; Beeson et al., 2017; Guerit et al., 2018). In these cases, drainage area adjustments in the form of divide mobility or river capture are thought to be occuring in response to the imbalance and will continue until equal values of χ exist across divides and planform network equilibrium is reached (Willett et al., 2014). Other recent studies suggest differences in χ across divides may better represent the potential for future divide mobility and may not always indicate active divide migration in the present (Whipple et al., 2017; Forte and Whipple, 2019).

We created batch $k_{sn}$ and χ maps (Fig. 5) given a minimum drainage area of 5 x $10^6$ m$^2$, a set 'smoothing distance' of 1000 m and a reference concavity of 0.5. For the main Awatere and Clarence rivers, we also used the TAK KsnProfiler functionality to make individual distance vs. elevation plots that show $k_{sn}$ and concavity values along sections of each river (Fig. 5b). These river segments were designated manually based on breaks in slope on χ-elevation plots (Perron and Royden, 2013) – see Supplementary Materials for individual plots (Fig. S2-S3). The $k_{sn}$ values for each of the Awatere and Clarence river segments are superimposed onto the batch steepness map shown as a heavier line weight (Fig. 5a).

## 4 Results: River and Fault Patterns

### 4.1 Planform Drainage Anomalies

In the western part of the MFS, north of the Hope fault, the upper Clarence river and its major tributary, the Acheron river, display numerous sharp bends (stars, Fig. 3a). In the same region, many of the tributary junctions are barbed and/or obtuse. The upper reaches of the Awatere river do not display sharp bends and barbed tributaries of the type observed along the Clarence river. However, the upper section of the Awatere river appears to be underfit, meaning the valley is wider than expected given the small drainage area. In the eastern MFS, north of the Hope fault, the Awatere river takes a relatively straight path northeast to the Pacific Ocean. In contrast, the Clarence river continues its irregular path, taking multiple sharp bends as

230 it flows first to the northeast between the Inland and Seaward Kaikōura ranges before turning sharply to the southeast to cut across the Seaward Kaikōura Range in a prominent water gap (green line, Fig. 3a). The river then makes two additional sharp turns over a short distance (~12 km), first to the southwest upon exiting the water gap, and then to the southeast to reach the Pacific Ocean (Fig. 3a). Similar, though smaller magnitude channel offsets, occur in many of the first order streams that flow across the Clarence and Awatere faults on the eastern range fronts. Several of these channels are barbed and obtuse (Fig. 3a)

235 and a few appear underfit. Drainage anomalies are less prevalent south of the Hope fault. In this region, there are no large, longitudinal rivers flowing parallel to faulted mountain fronts, nor do we find any barbed or obtuse tributaries (Fig. 2, 3a). A few prominent water gaps exist along rivers that flow through the Hundalee Hills to the Pacific Ocean (Fig. 3a).

## 4.2 Fault and River Orientations

240 Half-dial histogram plots with 10 bins° show orientations of rivers and faults from 0° (right/east) to 90° (center/north) to 180° (left/west), with circular mean orientations shown as thin black lines (Fig. 3b;4b). In Table 1, we report the circular mean and 1σ standard deviation of the measured river and fault orientations, as well as the value of the histogram bin that contains the highest percentage of segment orientations. Fault orientation results show that fault segments cluster within the eastern hemisphere of the dial plots (Fig. 4b), and that NW striking faults are rare in the MFS. Parsing the results by region shows that

245 the active and inactive fault orientations overlap within the eastern study domain, north of the Hope fault. In this region, most faults orient to ~40° (Fig 4b, middle panel; Table 1). Inactive and active faults south of the Hope fault also share similar orientations, with the majority of fault segments ranging from 30 – 40° and 40 – 50° for active and inactive faults, respectively. In contrast, active and inactive faults populate distinctly different orientation bins within the western domain, north of the Hope fault (Fig. 4b, left panel). Here, the older, inactive faults strike more northerly at ~55°, whereas the active faults strike

250 closer to east-west at ~30° (Table 1).

River orientations also show distinct patterns across the study area (Fig. 3). The majority of channels flow in a northeastern direction, though their orientations do not cluster as tightly in eastern hemisphere orientations as the faults (Fig. 3b). This scatter is reflected in river mean and standard deviation (Table 1) and likely arises in part due to analysis of streams of different

255 order and due to flow into the Pacific Ocean. The coastline through the field site is mainly NE (Fig. 1). As a result, even structurally-controlled channels must eventually flow to the SE to meet the ocean. Thus, channels with a more southern orientation show up in greater percentage in the eastern domain and south of the Hope fault, where the coastline and the ocean are located. Despite this dispersion, a pattern of overlap among river and fault orientations emerges (Fig. 3b). The majority of rivers in both the eastern domain, north of the Hope fault and the domain south of the Hope fault overlap with both active and

260 inactive fault orientations (Fig. 3b; Table 1). In the western MFS, north of the Hope fault, where active and inactive faults strike differently, the rivers tend to flow more northerly, overlapping in orientation closely with the inactive rather than the active faults (Fig. 4b, left panel; Table 1).

### 4.3 River Profiles: Channel Steepness and χ

Results of the channel profile analysis show high $k_{sn}$ values focused in two areas: 1) the northwest corner of the study area, in the upper Wairau catchment and 2) centered on the Kaikōura Ranges (Fig. 5a). Values of $k_{sn}$ drop off significantly south of the Awatere fault within the western MFS (Fig. 5a). South of the Hope fault, most rivers have $k_{sn}$ less than 150m. A few sections of river have higher $k_{sn}$ values, with steeper river segments along watergaps crossing the Hundalee Hills (Fig. 5a). The χ map shows a few instances of χ imbalances across divides, all located within the western MFS, north of the Hope fault (Fig. 5c).

### 5 Discussion

### 5.1 Interpretation of MFS River and Fault Patterns

We find an abundance of drainage anomalies in planform river patterns across the MFS, with the majority of features concentrated north of the Hope fault, in both the western and eastern portions of the study area (Fig. 3a). These features indicate recent or ongoing drainage rearrangement that affects the main Clarence, Awatere, and Wairau rivers and their tributaries. We suggest, based on ubiquity and proximity of drainage anomalies to faults and uplifting mountains, that disequilibrium in this drainage network relates largely to tectonic activity. The highest concentration of barbed and obtuse tributaries occurs in the western MFS, and indicates that rivers flowed north or west in the past before redirection to the south, possibly as a result of a series of captures by the headward extension of the main Clarence river along the Clarence fault (McCalpin, 1996, Craw et al., 2013) that would have redirected the Acheron, Saxton, and Alma Rivers. Within the same region, many tributaries to the upper Wairau river also exhibit barbed and obtuse forms indicating paleo flow directions to the south rather than to the modern-day course to the north. The underfit nature of the upper Awatere river may indicate that the river once had a larger upstream area that would have extended into the present-day upper Clarence watershed. A small water gap in the headwaters of the Awatere river (green line, Fig. 3a), along a tributary stream that also appears underfit, could be a previous pathway of the river, if indeed it once had larger headwaters to the west. A comparison of χ values across the study area also supports disruption to planform river patterns in the western MFS study region (Fig. 5c). This geomorphic domain shows a concentration of mappable χ anomalies, which may reflect present-day divide mobility or the possibility of future divide mobility in this region as a result of recent river captures.

In the eastern MFS, the main Clarence river flows parallel to the Kaikōura Ranges before cutting sharply to the east and south across the Seaward Kaikōura Range (Fig. 3a). This pattern suggests that the river flowed northeastward to bypass growing topography, eventually finding a way across the range at its lower, northern end, possibly by exploiting an antecedent stream. The last two bends appear to be in response to right lateral offset along the Kekerengu fault (Fig. 1), which slips at 24 (+/- 12)

mm/yr and has ruptured several times during the Holocene, including during the most recent 2016 Kaikōura earthquake (Little et al., 2018). Assuming a range in slip rate of 12 to 24 mm/yr and given a Clarence river offset of ~15 km, Kekerengu dextral

fault slip would have initiated between 0.6 and 1.2 Ma, a time range in agreement with the purported ~1 Ma onset of dextral motion on the adjacent, kinematically linked Hope fault (Wood et al., 1994; Van Dissen and Yates, 1991). Offset of the main Clarence river, as well as numerous smaller magnitude channel offsets within tributary channels flowing across the Clarence, Awatere, and Kekerengu faults correlate with regions of high concentration in drainage anomalies. These patterns show disequilibrium, likely as a result of the processes of lengthening and stream capture in response to local right-lateral motion

(Harbert et al., 2018).

Areas of high $k_{sn}$ also concentrate north of the Hope fault (Fig. 5a). Given the similar climate across the study area and that Torlesse graywacke rocks underlie most of the region (Rattenbury et al., 2006; Fig. S1), we interpret spatial variations in MFS river channel steepness indices in the context of spatially-variable uplift and river capture events. We recognize that sediment

flux, channel width, and changes in climate or geomorphic process through time may also affect the form of channel profiles. The area of high $k_{sn}$ in the northwest corner may be related to rock uplift associated with the restraining bend on the Alpine fault (Sagar, 2014; Harbert, 2019), which lies to the west, outside of the figure frame. This region was also glaciated in the past and the high channel steepness may reflect that.

The other zone of high $k_{sn}$ exists within the eastern geomorphic domain, stretching from the Awatere valley to the coast across the Inland and Seaward Kaikōura Ranges and the Papatea Block (Fig. 1). This zone of steep rivers overlaps with the highest topographic relief (Fig. 2;5a). We note, however, that the rivers display high $k_{sn}$ values even within the low-elevation main-stem river valleys. Thus, channel patterns do not simply reflect relief. High $k_{sn}$ here overlaps with the location of widespread, fast Pliocene-to-present exhumation determined from low-temperature thermochronology data (Collett et al., 2019). This area

also overlaps with a concentrated region of co-seismic vertical deformation that occurred during the 2016 earthquake (Hamling et al., 2017) and with a large part of the MFS underlain by the Pacific Plate (Eberhart-Phillips and Bannister, 2010). Other geomorphic features, such as strath terraces and bedrock exposed along the channels of the Awatere and Clarence rivers, including the stretch of the Clarence river just before it meets the ocean, and flights of marine terraces along the Kaikōura coast, all provide supporting evidence of rapid channel incision and high rates of rock uplift throughout the zone of high $k_{sn}$.

The spatial patterns in $k_{sn}$, topography and low-temperature thermochronology indicate an increase in transpression and rock uplift in the eastern MFS, toward a subduction front that is locked on its southern end (Little and Roberts, 1997; Walcott, 1998; Eberhart-Phillips and Bannister, 2010; Collett et al., 2019). Longitudinal profiles of the Awatere and Clarence support this hypothesis. Neither the Clarence nor the Awatere display typical concave-up graded profiles. Instead, the Awatere river profile

tends to be straight and the Clarence profile is slightly convex, with lower $k_{sn}$ upper reaches and higher $k_{sn}$ lower reaches (Fig. 5b), similar to what might be expected for detachment-limited rivers flowing from regions of lower to higher uplift rates

(Whipple and Tucker, 1999). The profile patterns could also relate to drainage capture events (e.g. Yanites et al., 2013; Seagran and Schoenbohm, 2019), such as those proposed to have occurred in the Awatere and Clarence headwaters based on planform patterns (Fig. 3a).

## 5.2 Controls on the Patterns of Drainage Networks at Long-Lived Tectonic Boundaries

The similarities between orientations of rivers and faults across the study area (Fig. 3b) reflects a strong structural control on the geometry of the drainage network consistent with the analysis of Roy et al. (2016a), which quantified topographic anisotropy across the MFS and suggested that drainage patterns reflect the fault network (see their Fig. 8). However, by parsing the active versus inactive faults in the different parts of the study site, we show that it takes more than simply breaking new faults to redirect drainage in long-lived tectonic settings. At present, rivers in the western MFS, north of the Hope fault, follow mainly the orientation of older, inactive faults, despite flowing orthogonally over strands of the active Awatere and Clarence strike-slip faults (see the upper Clarence and Acheron Rivers, Fig. 3a). This pattern in drainage likely reflects preferential flow along or adjacent to long-standing weaknesses along old faults that once accommodated the Cretaceous rifting of Gondwanaland. Rivers likely started flowing in this arrangement during the earliest parts of the Kaikōura Orogeny, when the landmass of the NE end of the South Island was emerging from the sea ($\geq$ 20 Mya).

The active, strike-slip faults through the western MFS domain are likely much younger than the inactive faults. Perhaps then they have not yet experienced enough displacement to produce significant fault damage (Savage and Brodsky, 2011) and thus may be too immature with respect to material-strength weakening to influence the strength field and engender drainage realignment. Field measurements of rock material properties from the Henry's Saddle section of the Fowlers fault, a subsidiary of the Awatere fault within the western MFS, show an ~3000 times decrease in cohesion in fault rocks as compared to surrounding intact rocks (Roy et al., 2016c). The authors estimate this difference in rock strength to correspond to an ~80 times increase in erodibility along the fault and note that the drainage in this area tends to follow this weak zone. These observations suggest that fault damage, at least on this section of the Fowlers fault, is enough to influence erosion and the drainage network. More comprehensive mapping and quantification of fault damage and bedrock erodibility across the Marlborough faults is required to fully assess the extent of weakening that has occurred with respect to the magnitude needed for widespread reorientation of rivers.

Though most of the rivers within the western MFS do not presently flow along the active faults, we suggest that strike-slip fault motion still plays a fundamental role in ongoing drainage network evolution. The few sections that presently flow along active fault segments are proximal to drainage anomalies indicative of recent capture and rerouting of the drainage. Laterally offset channels and off-fault deformation such as pull apart basins or pressure ridges likely act to disrupt established drainages (Duvall and Tucker, 2015), adding vulnerability and promoting capture of tributaries by the headward-eroding, eastward

flowing large rivers. Discrete river captures are a known driver of large-scale fluvial reorganization in orogenic belts (Struth et al., 2015; Babault et al., 2012). Thus, we suggest that strike-slip faulting, which promotes river capture, may be an integral ingredient to the ultimate rerouting of the river network away from the old fault network and into alignment with the active deformation field in this landscape.

Topography may also be an important factor to the development of a drainage network aligned along or parallel to the active faults. The ridge-valley topography generated by Miocene thrust-faulting in the Kaikōura Ranges is absent in the western MFS. There, the active faults are primarily strike-slip and have not generated the fault-parallel, high-relief ranges (Fig.1) that would aid in the development of longitudinal, along-strike, drainage of large rivers. In contrast, within the high-relief eastern MFS, the active and inactive faults are aligned, with ridges and rivers running parallel to the faults (Fig. 2). Notably, however, neither the main stem of the Clarence or Awatere rivers flows along their namesake active faults. Instead, the river valleys sit hundreds of meters below the faults, which are separated by as much as several kilometres in distance in some locations (Fig. 1). Tributary channels draining across these faults do manage to flow laterally for several kilometres along or near to the fault traces, but they all eventually carry on to meet the main river valley. The fault-parallel reaches rarely flow for more than the approximate spacing of range-perpendicular drainages before being shortened and rerouted by river capture from a neighbouring beheaded tributary (Harbert et al., 2018). One possibility is that main rivers originally flowed in fold valleys during the early stages of Kaikōura orogeny deformation, with the underlying thrust faults emerging later as deformation progressed (see Stage 1 in section 5.3 below). If the main rivers started flowing in the valleys before the faults daylighted at the surface, that would once again indicate the importance of entrenchment of a drainage network and the difficulty of rerouting drainages due to tectonic strain when not starting from a blank slate. Moreover, the issue of fault-damage extent and level of rock weakening could also be relevant here, depending on how recently the active strands became exposed at the surface and how wide of a damage zone has developed.

### 5.3 Landscape Evolution at the Boundary of Oblique Hikurangi Subduction and Continental Collision

In this section, we link large-scale drainage evolution to the known tectonic and deformation history of the MFS. Based on our landscape analysis, as well as previously published geologic and thermochronologic data, we propose two general stages of landscape evolution during the Kaikōura Orogeny: an initial relief-development stage related to thrust faulting and mountain building and 2) a channel-offset, headward-erosion, and river-capture stage from oblique dextral faulting and widespread uplift. We contend that both stages are recorded in the landscape. Below and in Figure 6, we describe the landscape evolution through four approximate time periods to explain the present-day geomorphic domains of the Marlborough region.

**Stage 1: Early to Late Miocene Thrust Faulting and Mountain Building** In our conceptual model, relief is generated early in the Kaikōura Orogeny by range-scale folds built above blind thrust faults within the Kaikōura domain (Van Dissen and

Yeats, 1991; Nicol and Van Dissen, 2002). Low-temperature thermochronology analysis suggests that the Inland Kaikōura Range began to form by ~ 25 Ma, and that both the Inland and Seaward Kaikōura Ranges were rising by 15 Ma (Collett et al., 2019). Although thermochronology data is not reported for the Awatere Mountains, we assume a similar, if not earlier development to the Inland Kaikōura Range (Fig. 6a). The ranges were likely built above reactivated structures, possibly
Cretaceous normal faults (Crampton et al., 1998; Nicol and Van Dissen, 2002). As the ranges developed, so too did the major longitudinal rivers, confined between adjacent highlands and flowing parallel to the range fronts in synclinal valleys (Fig. 6a; 6b). Drainage within the western Marlborough region was likely oriented north along inactive faults (Craw et al., 2013).

As deformation progressed into the mid-Miocene, we propose that the once blind reverse faults emerged to the surface, cutting
through the previously folded anticline limbs in a manner typical of sequential fold-thrust development (Berg, 1962; Brown, 1983). This model would be consistent with geologic mapping that shows the faults cutting across tilted strata (Rattenbury et al., 2006), and would explain why the faults are not within the major river valleys, at the low-elevation base of the mountain, but rather several hundred meters up the mountain front (Fig. 1;6b). During this time, Kaikōura Range relief would have increased as hanging wall uplift along reverse faults continued to build the mountains, and both the Awatere and Clarence
rivers incised as they responded to the increase in uplift (Fig. 6b).

**Stage 2: Late Miocene to Present Strike-Slip Faulting and Widespread Rock Uplift** By the late Miocene / early Pliocene, continued clockwise rotation of the Hikurangi subduction zone (Cande & Stock, 2004; King, 2000; Walcott 1998) led to a widening of the plate boundary and the onset of dextral faulting across the MFS (Lamb & Bibby, 1989; Knuepfer, 1992; Holt
& Haines, 1995; Little & Roberts, 1997; Little & Jones, 1998; Walcott, 1998; Hall et al., 2004). Around this time, we interpret the eastern sections of the main Marlborough faults to transition to oblique, primarily right-lateral strike-slip faults, from their earlier thrust or reverse sense of slip (Randall et al., 2011; Lamb, 2011). Rather than inheriting old structures, the strike-slip faults of the western MFS may have initiated new faults formed in orientations optimal to accommodate oblique plate convergence. Such a fault history would explain both the change in strike of the active faults by about 15° across the eastern
and western MFS (Lamb, 1988; Randall et al., 2011) and the differences in orientation between active and inactive faults in the western Marlborough region, north of the Hope fault, a trend not observed elsewhere across the study site (Fig. 6c).

In this latest part of the Kaikōura Orogeny, faulting and rock uplift became widespread in the eastern MFS, north of the Hope fault, including within low elevation river valleys and at the coast, in addition to continued uplift of the ranges (Collett et al.,
2019). Active fault segments with a slightly more northerly strike in the eastern MFS are more oblique to the plate motion vector and accommodate some contraction even as their primary motion is dextral (Van Dissen & Yeats, 1991; Little and Jones, 1998; Nicol and Van Dissen, 2002). In addition, dextral motion of crust along the Marlborough faults through the western MFS likely drives deformation and crustal thickening along eastern fault terminations. Seaward translation and overthrusting of crust atop the downgoing subducted slab (Little and Roberts, 1997; Walcott, 1998) is supported by geophysical

data that shows a zone of crustal thickening in the overlying plate (Eberhart-Phillips and Bannister, 2010). This pattern of widespread uplift is supported by the $k_{sn}$ map, which shows a hotspot of steep rivers across the Kaikōura Range highlands and lowlands (Fig. 5a).

During this second stage of landscape evolution, we propose that the combination of strike-slip fault motion along major faults, 430 regional uplift, and rapid incision of large rivers maintains relief in the Kaikōura Ranges and explains the triangular facets bounding the MFS faults in this region, as well as lateral offsets of rivers flowing over strike-slip faults (Fig. 6d). Recent numerical models of strike-slip systems (Duvall and Tucker, 2015; Harbert et al., 2018) with a similar set up of strike-slip faulting and regional uplift (equal uplift on both sides of the fault) show an ongoing cycle of stream-lengthening and capture for tributary channels that drain across strike-slip faults that produces landscapes similar to that of the modern MFS. The 435 modeled landscapes include horizontally offset channels, linear fault valleys running parallel to neighboring high topography, and high-relief steep hillslopes with faceted spurs adjacent to fault strands.

Expansion of the Awatere and Clarence faults and rivers westward during this time interval would have disrupted the western Marlborough drainage network. Indeed, studies of biological speciation of fish suggest that there was a connection between 440 rivers of the Marlborough region and the Canterbury region to the south until the late Miocene (Craw et al., 2016). In our conceptual model, strike-slip fault growth aided the headward erosion of major MFS rivers that first formed in the eastern portion of the study area, promoting drainage capture of rivers in the western MFS that flowed north (or south) along the old structural grain. We propose that enlargement of upper headwaters through piracy would have happened first along the Awatere River (Fig. 6c) and then along the Clarence River (Fig. 6d). Captures by the Clarence river included the southward deflection 445 of the Acheron, Saxton, and Alma rivers (McCalpin, 1996), a process that we suggest effectively beheaded the Awatere river (Fig. 6d). The underfit nature and low normalized channel steepness of the present-day upper Awatere river (Fig. 2;3) support the idea that it lost a once-greater headwaters.

Others have noted evidence of river captures and drainage reorganization in this region and suggested a history of Clarence 450 catchment drainage rearrangement as recently as the Quaternary (McCalpin 1992; 1996; Burridge et al., 2006). The convoluted flow path of the main Clarence river (Fig. 3a) suggests that growth may have happened through piecemeal captures and flow reversals of tributary drainages, likely aided by strike-slip faulting and the generation of local relief, as well as other factors, such as the opening of the Hanmer basin around 1 million years ago (Wood et al., 1994) and glacial erosion and deposition during cold phases of the Quaternary.

Southward migration of the Chatham Rise and the consequent propagation of the Hikurangi subduction zone and the overlying MFS (Little and Jones, 1998; Wallace et al., 2007) suggests that active deformation is presently focused at the southern end of the MFS, within the Hundalee Hills, and in the Porters Pass- Amberly zone to the south (Cowan et al., 1996). Continued uplift

on the Jordan-Kekerengu, Papatea, and offshore Kaikōura faults could lead to future mountains, water gaps, drainage deflections, and new longitudinal rivers (Fig. 6d). Uplift and fault activity south of the Hope fault are already generating topography, water gaps and some steep channels in the Hundalee Hills (Fig. 3a). We expect this landscape to evolve further as deformation becomes more focused in this region, possibly leading to through-going strike-slip faults (Cowan et al., 1996) and a hot zone of drainage anomalies and normalized channel steepness similar to the present Marlborough faults landscape.

## 6 Conclusions

Analysis of topography, fluvial morphologies in planform and profile, and comparison of orientations of faults and rivers, indicates that the Marlborough landscape evolved by drainage rearrangement from mountain uplift, strike-slip faulting and the headward erosion of main stem rivers during the Kaikōura Orogeny. This evolution includes faults in different orientations, some having been reactivated and others newly formed, and ultimately leading to disparate geomorphic domains across the MFS that reflect the various stages of orogeny experienced across the field site over the last 25 million years. Based on this tectonic and landscape history, we considered which factors dictated patterns in the drainage network over the lifespan of oblique convergence at this plate margin. Our dataset indicates the importance of fault characteristics such as age, displacement and sense of slip, as well as river characteristics, such as incision and entrenchment, headward erosion and river capture, in setting patterns in drainage networks, especially within transient landscapes at long-lived tectonic boundaries. We suggest that each of these elements must be considered when trying to glean information about past or present tectonic strain from orientation and patterns of drainage networks, particularly when multiple phases of orogeny are suspected.

## 7 Author contribution

AD performed the geomorphic analyses. AD prepared the manuscript with contributions from all coauthors. All authors participated in field work in the study area and connected field observations to the geomorphic analysis presented here.

## 8 Code and data availability

We used Topotoolbox and TAK codes in this paper, which are published and freely available (Schwanghart and Kuhn, 2010; Schwanghart and Scherler, 2014; Forte and Whipple, 2019). All data from this study appear in Table 1, the figures, and in the main text.

## 9 Acknowledgements

This work was made possible through generous support from the National Science Foundation (grants EAR-132859 and -1321735 to Duvall, Tucker, and Flowers, and -1126991 to Flowers). Upton was supported by the New Zealand Ministry for

Business Innovation and Employment (grant C05X1103). We appreciate the efforts of Wolfgang Schwanghart, Dirk Scherler, Philippe Steer, Adam Forte, and Kelin Whipple, whose Topotoolbox and TAK codes and routines were invaluable to this study. We appreciate valuable discussions with Alex Lechler, Philip Schoettle-Greene, Sean LaHusen, Seth Williams, and Erich Herzig that shaped the paper. Two anonymous reviews and comments from AE Jean Braun greatly improved the manuscript.


## Competing interests

The authors declare that they have no conflict of interest.

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

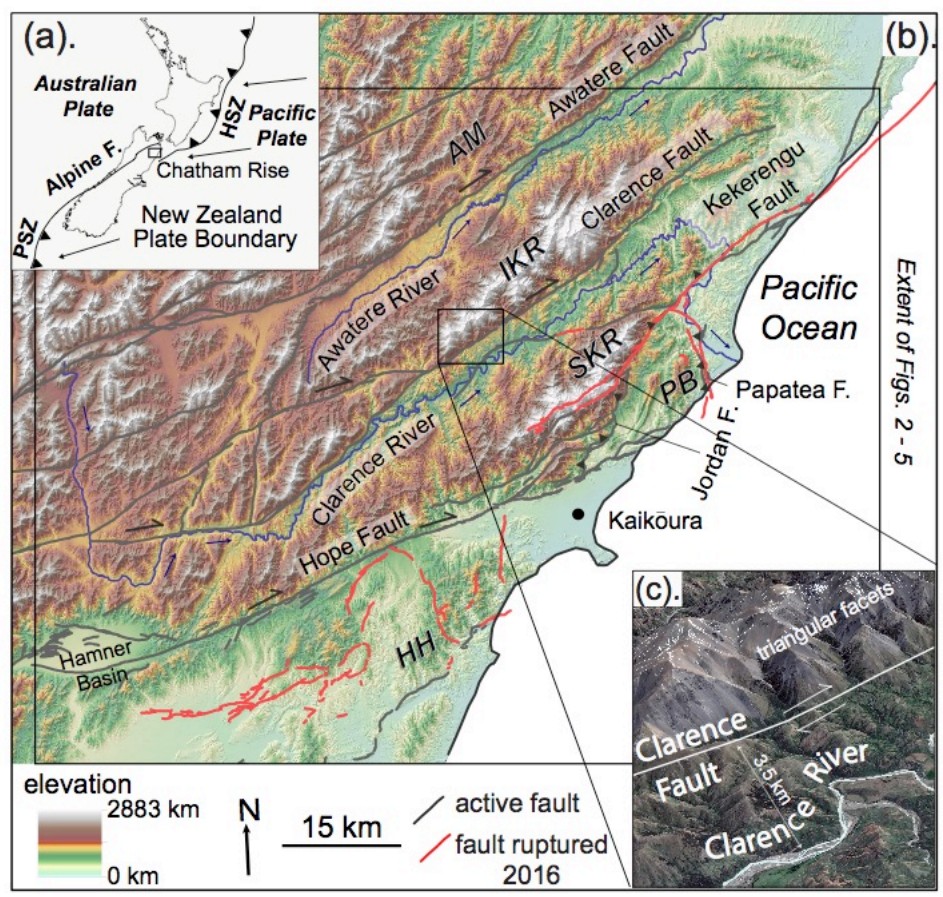

**Figure 1: Field Site Location Map: Marlborough Fault System, New Zealand. (a) Inset shows New Zealand plate boundary. HSZ – Hikurangi Subduction Zone, PSZ – Puysegur Subduction Zone. Arrows show plate boundary convergence vectors (De Mets et al., 2010). (b) Shaded relief map created from 8 m resolution digital elevation map from Land Information New Zealand. Main stem rivers shown in blue, with arrows indicating flow direction. Active Faults (black) and faults that ruptured during the 2016 Kaikōura**

**earthquake (red) from GNS Science fault database (Langridge et al., 2016). AM – Awatere Mountains, IKR – Inland Kaikōura Range, SKR – Seaward Kaikōura Range, PB – Papatea Block, HH – Hundalee Hills. (c) Blow up image of Clarence fault and Clarence river from Google Earth.**

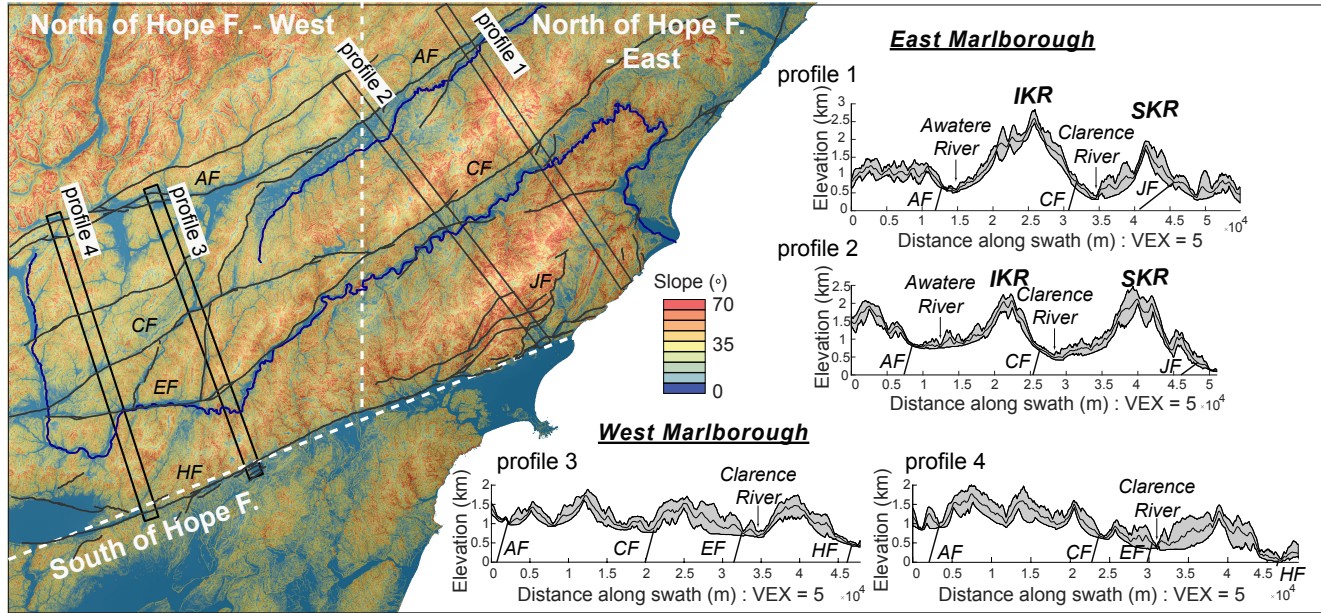

**Figure 2: (left) Slope map draped over shaded relief map of the study site. (right) Topographic swath profiles generated with the Topographic Analysis Kit (Forte and Whipple, 2019). Locations of profiles shown on slope map and labelled as profile 1, 2 (within eastern MFS) and 3,4 (within western MFS). White dashed lines show boundaries between 3 geomorphic domains based on differences in topography: western MFS, north of Hope fault, eastern MFS, north of Hope fault, and south of Hope fault. AF –**
**Awatere fault, CF – Clarence fault, EF – Elliott fault, JF – Jordan fault, HF – Hope fault.**

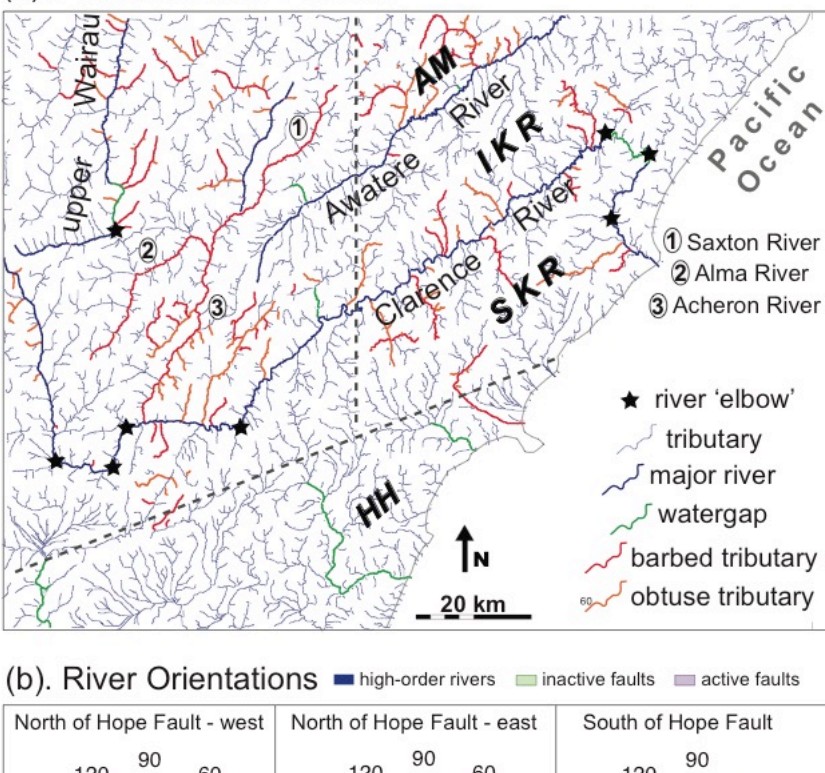

**Figure 3: Marlborough Fault System planform river patterns across three geomorphic domains. (a) Map of field site showing rivers and drainage anomalies. Main stem rivers shown in thicker line weight. AM – Awatere Mountains, IKR - Inland Kaikōura Range, SKR - Seaward Kaikōura Range, HH = Hundalee Hills. Black dashed lines show topographic domain boundaries (same as in Fig. 2) (b) Half dial plots showing orientations of river segments (blue) of Strahler order 2 and higher. Numbers within dials show the portion of segments analysed. Thin black line shows the circular mean. Each plot also shows the range in orientations of inactive (green) and active (purple) fault segments that represent the three histogram bins with the highest percentage of each feature (see Fig. 4).**

## (a). Planform Fault Patterns

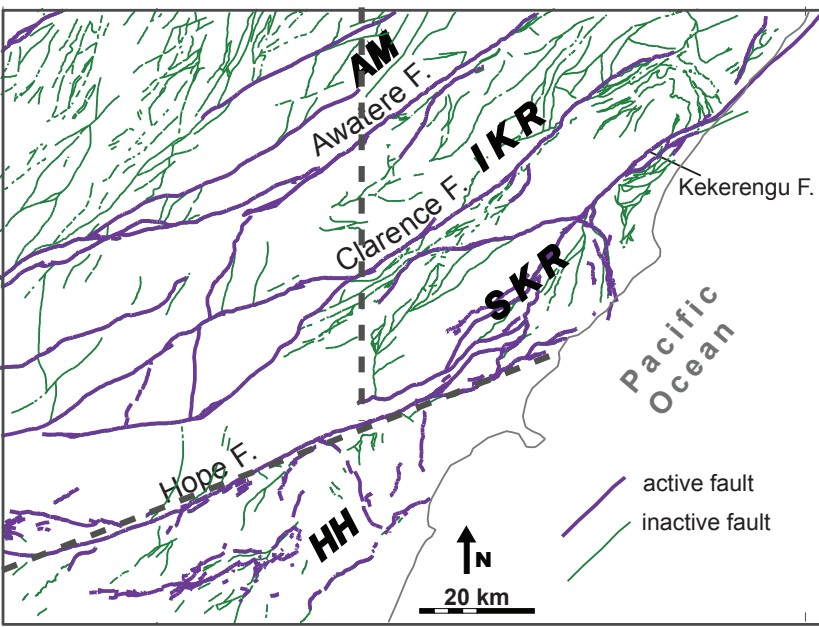

## (b). Faults Orientations

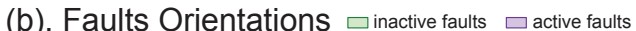

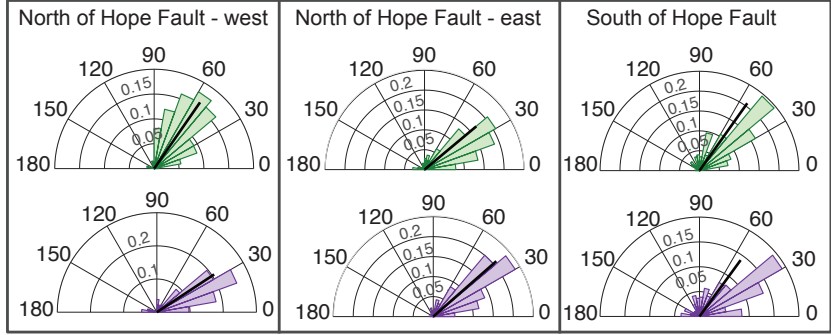

**Figure 4: Marlborough Fault System fault patterns across three geomorphic domains. (a) (a) Map of field site faults. Thicker line weight faults (purple) are active faults in GNS database. Thinner line weight faults (green) are inactive in GNS database. AM – Awatere Mountains, IKR - Inland Kaikōura Range, SKR - Seaward Kaikōura Range, HH = Hundalee Hills. Black dashed lines show topographic domain boundaries (same as in Fig. 2) (b) Half dial plots showing orientations of inactive (green) and active (purple) fault segments. Numbers within dials show the portion of segments analysed. Thin black line shows the circular mean.**

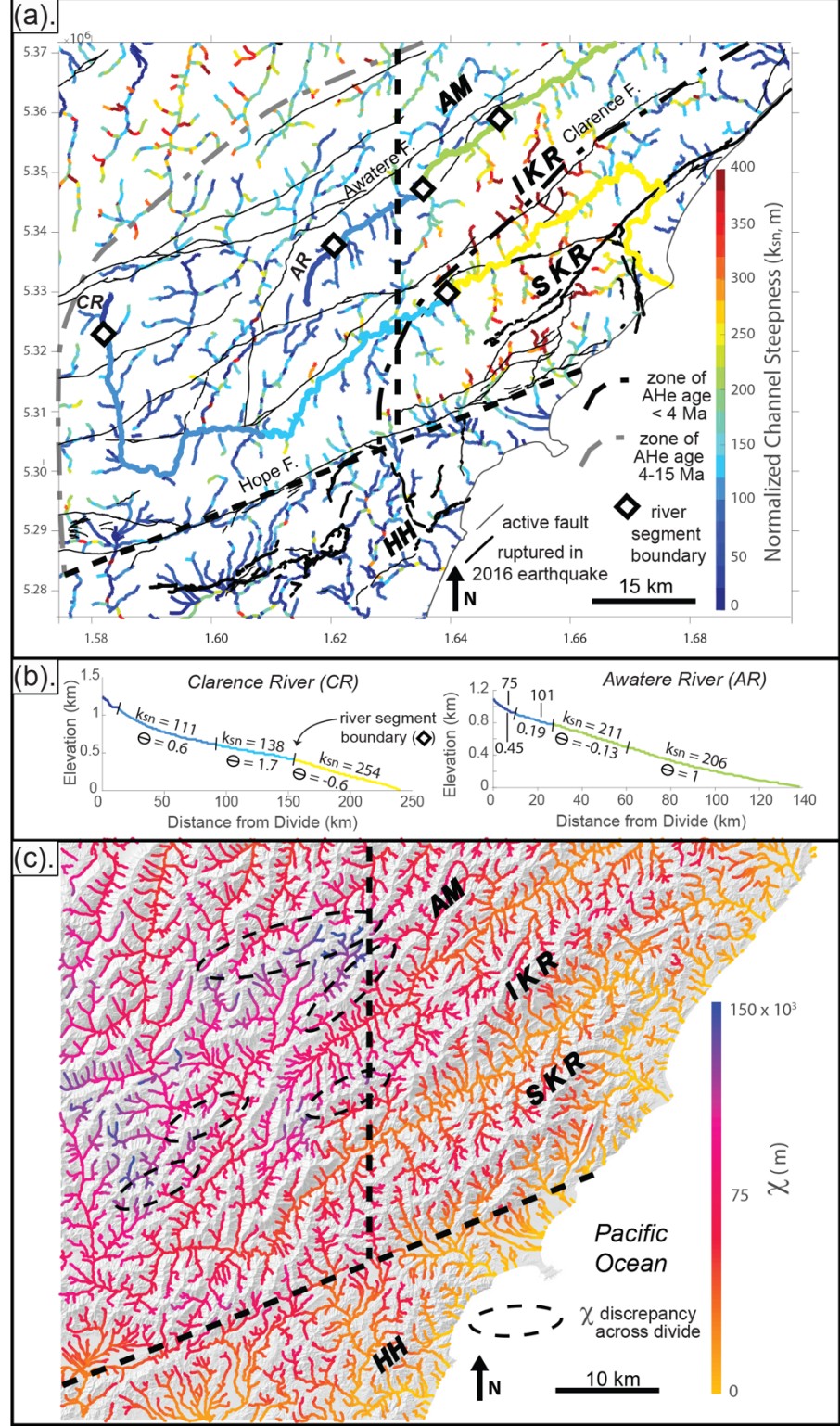

**Figure 5: (a) Map of normalized channel steepness for study site channels. Map generated using the $k_{sn}$ batch function in the Topographic Analysis Kit (Forte and Whipple, 2019). $K_{sn}$ of main Awatere (AR) and Clarence (CR) river segments (heavier line weight) and segment boundaries (white diamonds) estimated from manual designation of segments on longitudinal channel profiles using the $k_{sn}$ profiler function (see - b). Black and gray dot-dash polygons outline zones of AHe ages from Collett et al. (2019). AM – Awatere Mountains, IKR - Inland Kaikōura Range, SKR - Seaward Kaikōura Range, HH = Hundalee Hills. Black dashed lines show topographic domain boundaries (same as in Fig. 2) (b). Longitudinal river profiles of the Clarence and Awatere main stem rivers. Colors in b reflect the scale in a. $k_{sn}$ - normalized channel steepness, $\theta$ – concavity (c) Map of $\chi$ for study site channels generated using the Topographic Analysis Kit (Forte and Whipple, 2019). Black dashed polygons show location of $\chi$ discrepancies across divides.**

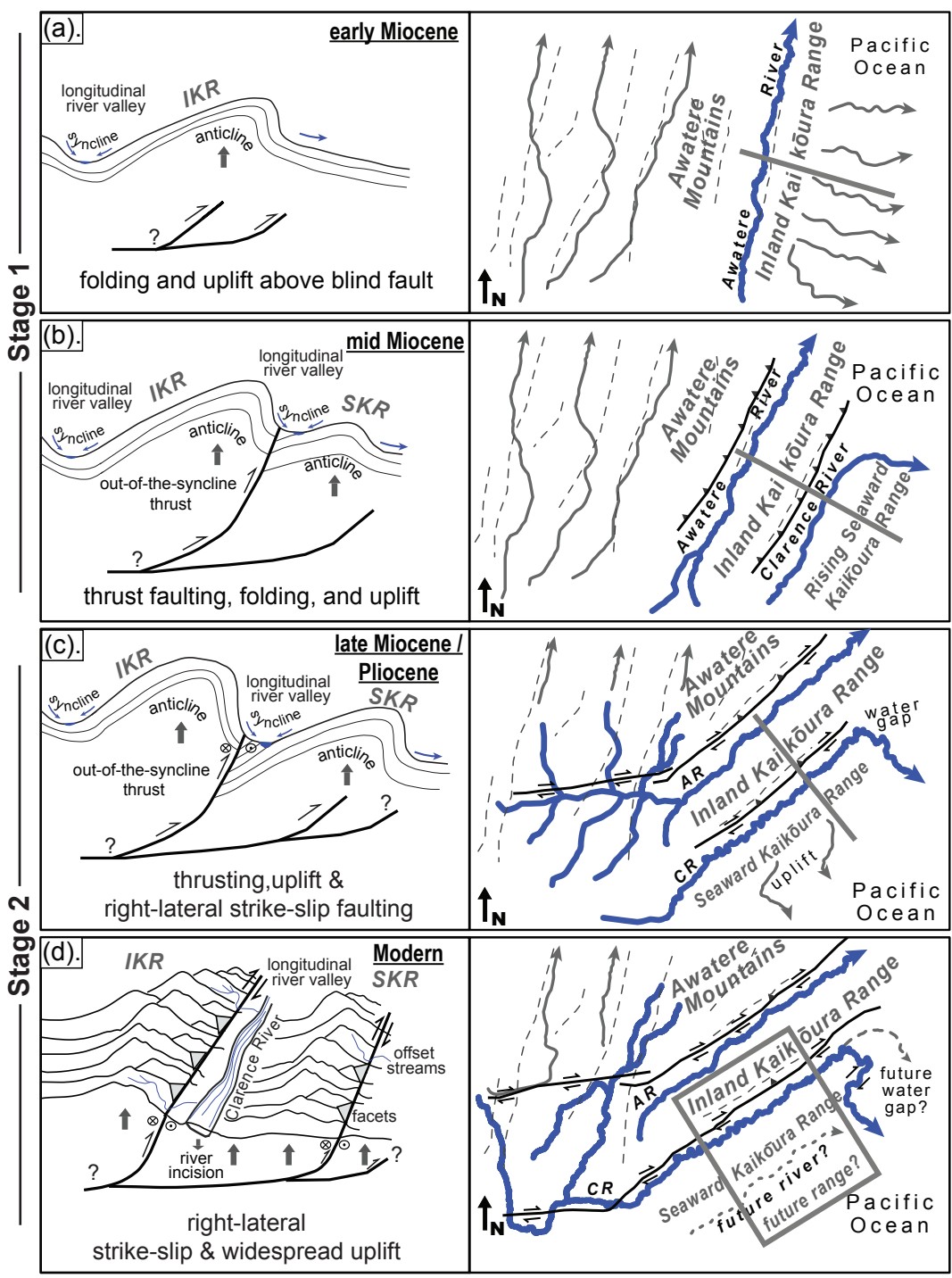

Figure 6: Schematic cartoon of Marlborough Fault System landscape evolution through time. Left panel shows cross-section view of the Kaikōura domain and right panel shows map view of landscape, expanded to include both the Kaikōura and Inland

**Marlborough domains. Approximate location of left-panel cross section shown with gray line (a,b,c) or box (d) in right panel. Right panel depicts active faults in black and inactive faults in gray and the approximate rotation of the faults and landscape from Randall et al. (2011). IKR - Inland Kaikōura Range, SKR - Seaward Kaikōura Range, AR – Awatere river, CR – Clarence river. See section 5.2 of text for a description of conceptual model.**

**Table 1: Fault and River Orientation Results**

| Domain Name | Inactive faults (highest bin) | Active faults (highest bin) | River (highest bin) | Inactive faults (mean/std.) | Active faults (mean/std.) | River (mean/std.) |
|---|---|---|---|---|---|---|
| N.Hope - west | 50° - 60° | 20° - 30° | 40° - 50°; 60° - 70° | 55°±12° | 33°±17° | 71°±33° |
| N.Hope - east | 30° - 40° | 30° - 40° | 40° - 50° | 40°±17° | 41°±14° | 75°±33° |
| S. Hope | 40° - 50° | 30° - 40° | 40° - 50° | 55°±14° | 54°±21° | 86°±22° |
