# Peer review of "River patterns reveal two stages of landscape evolution at an oblique convergent margin, Marlborough Fault System, New Zealand"

_Earth Surface Dynamics, 2019_

## Referee Comment (RC1) · Anonymous Referee #1 · 5 Sep 2019

This paper by Duvall et al is interesting, well written, well illustrated and well worth publication in eSurf.

**1   Response to eSurf's review guidelines.**

Does the paper address relevant scientific questions within the scope of ESurf?
**YES**
Does the paper present novel concepts, ideas, tools, or data?
**Data**

[Figure]

Are substantial conclusions reached?

**Partly regional (NZ northern tip), but with possible extension to other orogens.**

Are the scientific methods and assumptions valid and clearly outlined?

**Yes, but little criticism of the method (drainage anomalies) is presented.**

Are the results sufficient to support the interpretations and conclusions?

**Yes for the conclusion regarding the data and interpretations presented. But the general conclusion is also the starting point of the study.**

Is the description of experiments and calculations sufficiently complete and precise to allow their reproduction by fellow scientists (traceability of results)?

**Yes**

Do the authors give proper credit to related work and clearly indicate their own new/original contribution?

**They do give credit mostly to their own work, but omit a body of literature.**

Does the title clearly reflect the contents of the paper?

**Yes**

Does the abstract provide a concise and complete summary?

**Yes**

Is the overall presentation well structured and clear?

**Yes**

Is the language fluent and precise?

**Yes**

Are mathematical formulae, symbols, abbreviations, and units correctly defined and used?

**Yes**

Should any parts of the paper (text, formulae, figures, tables) be clarified, reduced, combined, or eliminated?

**Not really needed for the paper to reach its conclusions. But perhaps they could extend their review of previous work, their assessment of the limits of their method to 1) pick drainage anomalies and 2) interpret drainage anomalies, and**

**review if needed some of the more recent analysis related to chi-maps and its ability to reveal potential drainage rearrangement. Up to the authors.**
Are the number and quality of references appropriate?
**No. See 7.**
Is the amount and quality of supplementary material appropriate?

**2 General remarks/Questions**

I outline here below a few remarks that you may want to consider.

I am puzzled by the conclusion that "faulting, uplift, river capture and drainage network entrenchment all dictate drainage patterns and that each factor should be considered when assessing tectonic strain from landscapes, particularly at long-lived and complex tectonic boundaries". Indeed, this is the starting point on the research performed is based, as in the second part of the second sentence of the introduction: "Rivers, in particular, are influenced strongly by tectonic forces, as they are affected both by the ensuing mountain uplift (e.g.Whipple 35 and Tucker, 1999; Bishop, 2007) and by material weakening along faults (e.g.Koons, 1995; Molnar et al., 2007; Koons et al., 2012; Roy et al., 2015; 2016a;2016b;2016c)", and in the second chapter "Drainage network morphometry may additionally yield evidence of catchment reorganization of catchments by ridge migration (Pelletier, 2004; Willet et al., 2014) or by whole sale river capture (Craw et al., 2003; Clark et al., 2004; Craw et al., 2013) and flow reversal (Benowitz et al., 2019), often in response to tectonics. Recent studies focusing on fractures in bedrock channels have also revealed the importance of material strength in setting the orientation of rivers (Koons et al., 2012; Roy et al., 2015; 2016a; 2016b; 2016c; Scott and Wohl, 2019). Detailed field and numerical modeling studies of

rivers in faulted landscapes demonstrate the sensitivity of fluvial incision to gradients in erodibility between weak fault zones and the surrounding stronger bedrock (Roy et al., 2016c). These studies further show that structurally aligned drainages, with anomalously straight reaches, originate in response to the localization of fluvial erosion along the zones weakened by tectonic strain (Roy et al., 2015; 2016a;2016b)".

So, is there a circular argument here? Can you really conclude this if you start from it? Or should you rather say perhaps that you confirm your hypothesis? I do not know the answer, I just find it surprising to conclude on something which is stated as demonstrated in the introduction.

By the way, I think the paper by Molnar et al 2007 cited here is more on the influence of rock weakening on erosion in general, rather than on any influence on river patterns along faults.

I find that the way in which previous work has been reviewed, with regard to the questions posed, is a little too NZ-centered and ignores quite a significant body of literature. The papers you (self-)cite by Craw, Roy, Koons are all relevant (and acknowledged pioneer), but other people outside NZ have also worked on these questions. More than a sensitivity issue about the credit to these works (some of which is still measured by citations counts despite the DORA agreement), perhaps you may be interested in reading this literature. And there is much more than what I cite below.

The questions posed in the introduction "how do drainage networks evolve as an orogen deforms and over what timescales do the rivers respond to the changing tectonic boundary conditions? Are patterns in the drainage network overprinted as older faults change and new faults form? How important is the development of topography along

[Figure]

faults and the process of river capture in the establishment of structurally aligned drainages?" Have been addressed by many others, but I would advise reading some of the many papers by the group of Babault, e.g.:

Babault, J., Van Den Driessche, J., Teixell, A. (2012). Longitudinal to transverse drainage network evolution in the High Atlas (Morocco): The role of tectonics. Tectonics, 31(4), n/a–n/a. http://doi.org/10.1029/2011TC003015

Struth, L., Babault, J., Teixell, A. (2015). Drainage reorganization during mountain building in the river system of the Eastern Cordillera of the Colombian Andes. Geomorphology, 250(C), 370–383. http://doi.org/10.1016/j.geomorph.2015.09.012

Viaplana Muzas, M., Babault, J., Dominguez, S., Van Den Driessche, J., Legrand, X. (2015). Drainage network evolution and patterns of sedimentation in an experimental wedge. Tectonophysics, 664(C), 109–124. http://doi.org/10.1016/j.tecto.2015.09.007

Viaplana Muzas, M., Babault, J., Dominguez, S., Van Den Driessche, J., Legrand, X. (2018). Modelling of drainage dynamics influence on sediment routing system in a fold-and-thrust belt. Basin Research, 31(2), 290–310. http://doi.org/10.1111/bre.12321

And others as:

Giletycz, S., Loget, N., Chang, C. P., Mouthereau, F. (2015). Transient fluvial landscape and preservation of low-relief terrains in an emerging orogen: Example from Hengchun Peninsula, Taiwan. Geomorphology, 231(C), 169–181. http://doi.org/10.1016/j.geomorph.2014.11.026

Ramsey, L. A., Walker, R. T., Jackson, J. (2007). Geomorphic constraints on the active tectonics of southern Taiwan. Geophysical Journal International, 170(3), 1357–1372. http://doi.org/10.1111/j.1365-246X.2007.03444.x

Ramsey, L. A., Walker, R. T., Jackson, J. (2008). Fold evolution and drainage development in the Zagros mountains of Fars province, SE Iran. Basin Research, 20(1), 23–48. http://doi.org/10.1111/j.1365-2117.2007.00342.x

In the introduction you mention, "The planform rotation of river basins has been used as a marker of crustal strain (Hallet and Molnar, 2001) and to assess the style and rates of off-fault deformation (Goren et al., 2015; Gray et al., 2017)". Another interesting paper, also NZ-centered, but not cited, has demonstrated "assessing tectonic strain from landscapes, particularly at long-lived and complex tectonic boundaries" (from your abstract's conclusion) is by
Castelltort et al., in 2012, River drainage patterns in the New Zealand Alps primarily controlled by plate tectonic strain. Nature Geoscience, 5(10), 1–5. http://doi.org/10.1038/ngeo1582).

This group has also recently produced experimental tests of drainage networks as markers of stress or potential rearrangements (chi maps):
Guerit, L., Dominguez, S., Malavieille, J., Castelltort, S. (2016). Tectonophysics. Tectonophysics, 1–13. http://doi.org/10.1016/j.tecto.2016.04.016
Guerit, L., Goren, L., Dominguez, S., Malavieille, J., Castelltort, S. (2018). Landscape "stress" and reorganization from $\chi$-maps: Insights from experimental drainage networks in oblique collision setting. Earth Surface Processes and Landforms, 43(15), 3152–3163. http://doi.org/10.1002/esp.4477

Chapter 3.1
"Drainage anomalies, or unusual patterns in river planform, can indicate recent river captures and reorganization of drainage networks, often in response to active tectonics (e.g.Bishop, 1995; Brookfield, 1998; Hallet and Molnar, 2001; Burrato et al., 2003; Clark et al., 2004; Delcaillau et al., 2006; Willett et al., 2014)". Since you cite Bishop 1995 here, who actually provides an in-depth examination of this issue, I would emphasise that the "can" is very important: indeed in the next sentence you explain "We mapped drainage anomalies across the study site and found abundant evidence of fluvial disruption within both domains. Following McCalpin (1996) and

[Figure]

Craw and Waters (2007), we demarcated river elbows, locations where major rivers take an 90° bend and barbed tributaries, which are channels that join their main river in an upstream rather than downstream direction (Fig. 2b)" . In the rest of the chapter describing these observations, the interpretation provided is that these anomalies "likely, possibly, could" indicate e.g. captures, rearrangements etc. => So, how robust is the interpretation of such "anomalies"?

"In the earliest phase of the KaikÅ■ura orogeny": hard for outsiders to know when that is, perhaps it would be good to put xMa in brackets after this and elsewhere in the text (rifting of Gondwanaland and/or early KaikÅ■ura orogeny shear).

"There, the active faults are primarily strike-slip and have not generated the fault-parallel, high-relief ranges (Fig.1) that would aide in the development of transverse drainage" - It can be readily observed in many mountain ranges, but also in field and roadcuts, or in the lab, or in numerical experiments, that transverse drainage develops easily, without needing the aide of faults. See Hovius 1996 for instance for a first review of this.

Hovius, N. (1996). Regular spacing of drainage outlets from linear mountain belts. Basin Research, 8, 29–44.

The rest of the chapters describes and quantifies to some extent drainage in the study area and examine the relation between drainage and the tectonic pattern. I regret a lack of use of recently utilised chi-maps (see papers by Willett's group for instance: Willett, S. D., McCoy, S. W., Perron, J. T., Goren, L., Chen, C. Y. (2014). Dynamic Reorganization of River Basins. Science, 343(6175), 1248765–1248765. http://doi.org/10.1126/science.1248765

Fox, M., Goren, L., May, D. A., Willett, S. D. (2014). Inversion of fluvial channels for paleorock uplift rates in Taiwan. Journal of Geophysical Research: Earth Surface, 119(9), 1853–1875. http://doi.org/10.1002/(ISSN)2169-9011)

I would say that "correlation does not imply causation", and some statistical tests of the means / medians could add the comparison, but still, here the correlation is very convincing and I think the conclusions of the authors that "our dataset indicates the importance of fault characteristics such as age, displacement and sense of slip, as well as river characteristics, such as incision and entrenchment, headward erosion and capture, in setting patterns in drainage networks" is well supported by the data and interpretations.

Last remark: what do you mean by "mature"?

**3 Bibliography**

Babault, J., Van Den Driessche, J., Teixell, A. (2012). Longitudinal to transverse drainage network evolution in the High Atlas (Morocco): The role of tectonics. Tectonics, 31(4), n/a–n/a. http://doi.org/10.1029/2011TC003015 Fox, M., Goren, L., May, D. A., Willett, S. D. (2014). Inversion of fluvial channels for paleorock uplift rates in Taiwan. Journal of Geophysical Research: Earth Surface, 119(9), 1853–1875. http://doi.org/10.1002/(ISSN)2169-9011) Giletycz, S., Loget, N., Chang, C. P., Mouthereau, F. (2015). Transient fluvial landscape and preservation of low-relief terrains in an emerging orogen: Example from Hengchun Peninsula, Taiwan. Geomorphology, 231(C), 169–181. http://doi.org/10.1016/j.geomorph.2014.11.026 Guerit, L., Dominguez, S., Malavieille, J., Castelltort, S. (2016). Tectonophysics. Tectonophysics, 1–13. http://doi.org/10.1016/j.tecto.2016.04.016 and Guerit, L., Goren, L., Dominguez, S., Malavieille, J., Castelltort, S. (2018). Landscape "stress" and re-organization from $\chi$-maps: Insights from experimental drainage networks in oblique collision setting. Earth Surface Processes and Landforms, 43(15), 3152–3163. http://doi.org/10.1002/esp.4477 Hovius, N. (1996). Regular spacing of drainage outlets from linear mountain belts. Basin Research, 8, 29–44. Ramsey, L. A., Walker, R. T., Jackson, J. (2007). Geomorphic constraints on the active tectonics of southern Taiwan. Geophysical Journal International, 170(3), 1357–1372. http://doi.org/10.1111/j.1365-246X.2007.03444.x Ramsey, L. A., Walker, R. T., Jackson, J. (2008). Fold evolution and drainage development in the Zagros mountains of Fars province, SE Iran. Basin Research, 20(1), 23–48. http://doi.org/10.1111/j.1365-2117.2007.00342.x Struth, L., Babault, J., Teixell, A. (2015). Drainage reorganization during mountain building in the river system of the Eastern Cordillera of the Colombian Andes. Geomorphology, 250(C), 370–383. http://doi.org/10.1016/j.geomorph.2015.09.012 Viaplana Muzas, M., Babault, J., Dominguez, S., Van Den Driessche, J., Legrand, X. (2015). Drainage network evolution and patterns of sedimentation in an experimental wedge. Tectonophysics, 664(C), 109–124. http://doi.org/10.1016/j.tecto.2015.09.007 Viaplana Muzas, M., Babault, J., Dominguez, S., Van Den Driessche, J., Legrand, X. (2018). Modelling of drainage dynamics influence on sediment routing system in a fold-and-thrust belt. Basin Research, 31(2), 290–310. http://doi.org/10.1111/bre.12321 Willett, S. D., McCoy, S. W., Perron, J. T., Goren, L., Chen, C. Y. (2014). Dynamic Reorganization of River Basins. Science, 343(6175), 1248765–1248765. http://doi.org/10.1126/science.1248765

---

## Referee Comment (RC2) · Anonymous Referee #2 · 5 Sep 2019

This paper presents topographic and geomorphic data to evaluate the tectonics and associated drainage systems evolution in a complex tectonic setting. The data and analyses presented are valuable and well worth publishing in ESurf. However, I think that for this paper to reach its full potential, and the widest audience possible, significant changes are needed. Most of my detailed comments, below, address the three following main issues: (1) the paper needs a Methods sections, clarifying the techniques and criteria followed for their analysis; and also a much greater separation of the methods, results and discussion, as well as greater differentiation of which are the interpretations derived from their analysis and from published data. References to published data and interpretations are spread across all sections of the paper, making it

hard sometimes to let this paper's findings come through. (2) The paper often relies too heavily on readers being very familiar with other papers, either on the methods, or on the background geological setting and previous studies in the area. By down so, the authors are narrowing down their readership, making it more regional- and expert-focused. Providing a wider background on the previous geologic constraints, and more explicit information about the methods would attract a broader readership interested in drainage evolution linked to tectonics, but not familiar with the area, or interested in the study area but not familiar with drainage analysis. (3) Is it really necessary to separate the study area in two "domains"? Given that the divide is arbitrary and that many geomorphic and tectonic characteristics are transitional, this does not seem necessary, and in some cases, doing so unnecessarily complicates the analyses (see comments below). I suggest just referring to the ENE and WSW sides of the study area, or using some features (town names, peaks, etc.) as references.

Introduction

Line 50: based on your later results, I suggest writing "the position and orientation of rivers" rather than only orientation.

Lines 66-67: in this sentence you are listing all your analysis, so I do not think that saying "including" is appropriate here, as it gives the impression that there are more analysis than those on the list. I would rephrase to simply say "In this study, we present analysis on the topography, fluvial morphologies in planform and profile forms, and orientations of rivers compared to active and inactive faults"

Geologic Setting

At present, the last 3 paragraphs of the Geologic Setting read a bit convoluted because they go from making a general statement on the overall evolution, to talking about the present-day configuration and slip rates, to the early deformation phase, and the evolution from Late Miocene to today. I would suggest following a chronological order, so switching lines 88-94 to the end of the section. Alternatively, if the authors prefer presenting the names of the faults before talking about the geological evolution, I suggest these lines should go in the 1st paragraph of the section, so that once the geological evolution starts to be discussed, a clear chronological order is followed. Lines 88-92 could go after the "The MFS. . . collision" sentence, and lines 92-94 could go at the end of the paragraph, so that discussion about the 2016 earthquake is not spread across the ends of two different paragraphs.

Line 90: how have these slip rate estimates been derived? GPS? Offset dated surfaces? A large number of studies are referred, but readers should not need to be familiar with those in order to have a general idea – a general statement saying "derived from. . ." would be helpful.

Line 97: please be more specific with the geologic time you are referring to when saying "Early in the plate boundary history" (is it Late Oligocene, Early Miocene, Early to Mid Miocene. . .?). You could add a parenthesis specifying this before the coma.

Line 98: what type of structures? Just saying "a few important structures" is vague. Figure 4 suggests that these were primarily thrusts and folds associated with them, but this information should be clearly presented in the geological setting, particularly given that it is going to be heavily included in the discussion.

Line 104: again, I think the readers would benefit from greater clarity on the time you are referring to (25 Ma?). Also, to follow a clear chronological order, I would suggest that this sentence goes when the geological history is starting to be discussed, at the beginning of the 2nd paragraph.

Line 108: here or when discussing current slip rates – could you provide with some estimates on the partitioning of vertical vs. lateral motion? "Lesser" is quite vague.

Line 110: readers would benefit from a brief statement describing how have the "estimates of timing, cumulative decrease in total offset, and increase in slip rates" have been derived, or at least what type of data set they come from. Also, could you please

explain what is meant by "cumulative decrease in total offset"? I understand how an increase in offsets could inform about the time since fault activity started, but I am not sure how could a decrease in offsets inform of that, or how it could even be identified or resolved.

Topography and planform river patterns

Given that the dividing line is arbitrary, and that many of the landscape features are transitional – is dividing the área in "domains" actually necessary? I suggest that the authors simply refer to the ENE and WSW parts of the study area, or include some other features (peaks, towns) as a point of reference, rather than making an arbitrary division that I also think complicates their interpretations in the following figures, given that this divide position does not actually correspond with any clear geomorphic boundaries.

The first two paragraphs of this section read a lot like information that should be on the Geological Setting section, given that, except the swath profiles, there is no "result" of analysis presented. I suggest moving these paragraphs that simply describe the landscape features from the DEM and the positions of the faults based on published data, to a sub-section of the Geological Setting. Also, I see that the DEM used and the source of the faults map are listed in the caption of the figure, but this is important information that should be included on the main text, in a Methods section. Drainage anomalies

Lines 138-139 belong in the introduction, along more background information about the use of these anomalies to infer tectonic perturbations. Why are these particular features chosen for analysis, and what are they indicative of? Rather than simply saying here "Following McCalpin (1996) and Craw and Waters (2007)", I suggest briefly summarizing these previous works in the introduction, and why river elbows, barbed tributaries, etc. are can be indicative of drainage perturbations related to tectonics. Before presenting the results, a description of the criteria followed to identify an "anomaly"

should be presented in a Methods section, which should include information on how have river elbows, barbed tributaries, water gaps and underfit channels have been defined and identified. Why have river elbows only been marked in the main channels? How do the authors assess if a channel is "fit" or "underfit", have they used any published (or compiled themselves) graph of valley width vs. discharge? Do the channels mapped as "underfit" deviate sufficiently from the overall trend to be distinctly identified? A graph showing the overall trend of valley width vs. discharge, and how "underfit" channels deviate should be included in the main text or in the supplementary information.

Information on the maps used and their resolution also belongs in a Methods section, not in the results.

Lines 154: can you use these slip rate estimates to infer minimum time since the Clarence started flowing to the SE in its lower reaches? Or using the offset to estimate the beginning of slip in the Kekerengu fault? This would better highlight the potential of drainage patterns on informing about tectonic evolution.

Line 166: the previous line mentions both the Awatere and the Clarence river, so it is not clear what river and what segment is referred to when saying "this segment", please be more specific.

Lines 167-168: this short sentence says twice "in the headwaters of the Awatere river" – I suggest rephrasing to "In the headwaters of the Awatere river, a small water gap and an underfit stream (number X and X on figure X) could indicate the previous pathway of the river, if indeed it once had larger headwaters to the west".

Figure 2a and 2b: these are two important figures for the paper's results, but it is often hard to follow the results because the figures are too small and cluttered, and two important features for the analysis, the relief and the faults, are displayed in other figures, making it harder to relate them to the drainage network. I suggest moving the faults and river orientation analysis (panels c and d) to another figure, and make this

figure a bigger panel figure with 4 or 2 panels, each showing the drainage network, the faults, and hillshade relief in black and white (or the DEM in a paler color scale), but in each highlighting in color only one or two of the mapped features (watergaps, elbows, underfit channels, barbed tributaries). This would also allow more space to add Id or labels to the key features discussed in the text, so that rather than saying "a small water gap in the headwaters (e.g. line 167)", the authors can write "a small water gap (n°14) in the headwaters". This would considerably help following the information presented in this section. Also, the orange and red colors chosen for underfit rivers and barbed tributaries are very hard to differentiate in printed versions of the paper, I suggest displaying these in different panels or using a more different color for one of them.

Orientations of Rivers and Faults

Lines 175-177: These sentences belong in the Geological Setting, they are not the results of this paper. The "every-direction variogram" analysis is not routinely used in geomorphology, so the authors' analysis is going to build up on this, they should include a methods section in which they summarize the method.

Lines 180-183: These information is important, but belongs in a methods section. Also, please explicitly state whether you follow the same criteria as GNS to consider if a fault is active or inactive, and what do you mean by "mature" faults (ie. An inactive fault could be mature? For example if it was active for long enough to significantly weaken the bedrock).

Clearly A5 and A8 span both domains, so it is problematic to overlap the previous Inland Malborough vs. Kaikoura domains to this grid pattern. I understand the practicalities of diving the area in grids, but as they are right now, these grids are not truly representative of the different areas, and if anything, they could be masking some trends. The choice of number and size of grids should also be discussed in the methods, as right now it seems highly arbitrary, and that it can have a strong impact in the

results presented in Fig.2c and 2d. Is it not possible to rotate the grid or map to align the grid boundaries with the dominant ENE-WSW pattern of the faults and the relief? That way the different grids would be more representative of the true gradients in relief and tectonics (rivers and faults should still show in the radial plots their true orientation, I just suggest changing the reference grids used to divide the area up in zones).

Line 186: why have these channel orders been selected? This needs a brief justification in the methods.

Line 187: some description of what these "network segment and plotting routines" are and do is needed in the methods. Readers should not have to be familiar with Philip Steer's contribution to TopoToolbox in order to at least have a 1st order idea on how you have treated your data in your paper. Of course you can always redirect readers to published studies for more detailed information, but the core of the analysis and methods should be briefly discussed. If no paper exists for these contributions, please provide a link as reference.

Line 186: again, this belongs in the methods, it is not a result. How have you done this normalization? Can this normalization mask results, if the orientation of the largest, dominant faults means that their influence in the graphs is heavily weighted down by their length?

I strongly suggest using a more quantitative, statistics-based way to assess the overlap of the inactive and active faults and river orientations (Also, perhaps adding a box plot near the radial plots could help visualize the overlapping better?). Looking at panels A2 or A3 for example, I would never say that the orientations "overlap strongly" as it's said in Line 191. Even visually, it is hard to fully assess the overlapping when the active faults are depicted in opaque black (also, maybe two different translucent colors could be used for active and inactive faults, so that overlapping areas can be more easily visualized as a color combination?).

Line 192: what about the fact that in A1, A2 and A3 active faults have a much narrower

distribution than inactive ones?

Line 193: it is hard to assess the true degree of overlapping for A7 (translucent colors may help, see above), but A4 appears to have a significant number of overlapping rather than two "clustered" different populations as it is mentioned here.

Line 195: it is a bit bold to say that all the difference between active and inactive faults orientation comes from the influence of the Malborough domain, looking at the data this does not seem to be the case. I would suggest for example using a different color scheme for this "all faults plot" (and perhaps making it bigger, showing it as another entire panel), with different colors for each area, but different degrees of opacity for active/inactive? Also, maybe you could you use the "degrees of rotation" from the inactive to active faults population to reconcile it with the overall rotation estimated for the deformation field from previous studies?

Line 200: do rivers in the Malborough align N-S? it seems to me that a NE orientation prevails, which is pretty similar to many rivers in the Kaikoura domain.

Lines 202-203: A4 and A7 in the Malborough domain have almost as many NW-SE orientated rivers as A8 has.

Overall I find the results and discussion of this section unclear, because it relies on qualitative and subjective visual assessments and on the overlapping of two arbitrary sets of divisions on the study area, which mask the important key findings: a) active faults are more E-trending than inactive faults, (b) overall, river and fault orientations overlap, (c) to the NE and S of the study area, there are also NW-SE-oriented rivers that do not overlap with the existing faults. Please see my comment at the beginning of this review about the Inland Malborough vs. Kaikoura domain separation, I suggest eliminating this arbitrary separation, even more on light of the results presented in Fig.2c and 2d. This would also make the presentation of results more straightforward and focused on the overall, significant trends.

River Profiles and Channel Steepness

Lines 208-218: none of this are results from this study, this paragraph belongs in the introduction.

Line 220: I suggest adding a lithological map of the study area, it would be very helpful for readers not familiar with this area of NZ but interested in your drainage evolution results.

Lines 225-234: all this belongs in a Methods section. What "default values"? From what paper/software? Rather than simply saying "we use other default values", please include all parameters used in a table on the supplementary information. Why is this particular drainage area threshold used?

Line 233: the chi-plots used to identify breaks in slope should be included in the supplementary information.

Line 236: ksn should have units of m0.9 if a reference concavity of 0.45 ($\sim$0.5) is used.

Lines 245-251 and 254-256: contextualizing this paper's findings with previous published studies belongs in the discussion, not in the results. Also, if discussion is going to refer often to several published thermochronology studies, I would suggest adding a summary figure with the available thermochronological data and the exhumation patterns derived from this (could be another panel in Fig. 1 for example).

Line 58: the ksn value presented in Fig. 3b is actually lower for the lower Awatere reach than for the intermediate one. . .

Landscape evolution at the edge of the Hikurangi subduction

I would strongly suggest discussing the key findings and interpretation of your data analyses first – i.e. what do they indicate in terms of drainage evolution, and what type of tectonic perturbations would they suggest? – before contextualizing your data in the wider geological setting. Essentially switching the order of your current sections

5 and 6. You say in line 269 that you link the large-scale drainage evolution with the known tectonic history, but for that, a summary of the large-scale drainage evolution that includes the key findings and interpretations of your data should be provided first.

In the text, you write "stage 1, stage 2. . ." but in the figures you write "Early Miocene, Mid Miocene. . .". Please be consistent so that it is easier to follow. For example, you could write I both: "Stage 1: Early to Mid Miocene", "Stage 2: Mid to Late Miocene", etc.

Line 292: and as they responded to the increase in uplift. . .

Line 332-333: this should have been mentioned in the results.

Discussion

Line 364: I suggest adding "enough displacement to [. . .] or to produce significant relief"

Conclusions

Line 400: please do state explicitly what factors were investigated – many people read the conclusions of a paper before deciding whether to read it entirely or not, so this would be relevant information.

TYPOS, ETC.

Line 38: space missing between "e.g." and "Wobus" Line 17: for clarity, please insert "drainage" here, so that it reads "history of drainage capture and rearrangement" Line 64: space missing between "e.g." and "King" Line 69: I suggest changing "complicated" for "complex", otherwise the word "complicated" is repeated 3 times in 6 lines. Line 75: typo, "Puysegur" not "Puyseguer" Line 139: space missing between "e.g." and "Bishop" Line 409: it is "Philippe Steer" not "Phillipe Steere"

---

## Editor Comment (EC1) · Jean Braun (Editor) · 6 Sep 2019

Reading the two reviews posted so far about this manuscript, I find them very constructive and useful for the authors if they wish to start preparing a revised version. Both reviewers acknowledge that this is an interesting manuscript but that it requires to be improved before publication in ESurf.

In particular, there is a need to expand the description of previous work and its comparison with the results obtained in this paper. This would ensure that the paper is appealing to a broader audience. I thank one of the reviewers for taking the time to provide a very thorough account of previously published work on the tectonic control of

river network geometry.

One of the reviewers also find that the paper needs to be better organized with some of the material describing previous work and the geological setting currently dispersed in various parts of the manuscript needing to be moved in the introduction or just below. Both reviewers also express the concern that it is not clear what is a conclusion or an assumption (similar statement s appear in the introduction and the conclusion/discussion sections).

I also agree with one of the reviewers that the methodology used must be greatly improved. As it is, I do not believe that the results obtained here could be easily reproduced with the information provided by the authors.

However, I do not wish to appear too negative about this paper. It reports very interesting results that should definitely be published in ESurf but requires to be presented in a way that makes it of greater general applicability/relevance and is better organized.

The discussion will remain open until the 18th of December in the hope of attracting additional comments and/or criticisms.

---

## Author Comment (AC1) · 9 Oct 2019

We appreciate the thoughtful, thorough, and constructive responses from the reviewers and the associated editor. Both referees brought up several valid concerns and provide many excellent suggested edits and comments to address these. We intend to revise the manuscript in accordance with these suggestions (see comments to each below) and feel the paper will be improved significantly as a result.

Anonymous Referee #1

Referee #1 noted that we focused our background and discussion too narrowly on New

[Figure]

Zealand and as a result omitted a large body of literature on the subject of drainage network evolution in faulted landscapes and with respect to material strength heterogeneities in bedrock. We appreciate their additional literature suggestions and plan to incorporate these and other relevant works into the revised manuscript. This should provide a more complete treatment of the subjects in question and help to make our paper more universally relevant.

They also pointed out the need for more method details and a distinct methods section of the paper. We plan to write a more complete methods section in a revised manuscript. Also, with respect to methods, they suggest a statistical analysis of the orientation data and the addition of a chi map. We agree that these would be valuable additions and will include both in a revised version.

Referee #1 also finds that the way the manuscript was presented came across as somewhat circular and suggest that we more clearly state the hypotheses that we are testing upfront and revisit these in the conclusions. We plan to revise the manuscript writing with this in mind.

Anonymous Referee #2

Referee #2 echoes some of the comments by Referee #1 and adds other important concerns.

We intend to revise the paper format to provide a more appropriate differentiation of background, methods, results, and interpretation. We regret that the original version was unfocused and difficult to follow. We are also expanding the background as well as adding more complete descriptions of the analysis in a methods section, including statistics on the orientation data and adding chi map (see comments to Referee #1 above). We will also reorder the discussion section to begin with a discussion of the key findings and interpretation of our data before placing this in the context of the broader geologic setting.

We agree that our original 8 subsections (A1 – A8) of the field site for fault and river orientation analysis were arbitrary and as a result, might be obscuring or unnecessarily confusing our results. We have redone the analysis by dividing the landscape into three separate sections (NE, SW, and South of the Hope Fault).

We plan to separate sections of figures so as to avoid clutter and emphasize the importance of the different analyses. For example, we plan to separate the fault orientations and river orientations results into two separate figures.

Referee #2 also provided numerous line edits, questions, comments and we intend to address each one in the revised manuscript.

---

## Author Response (AR2)

Response to Reviewers/Editors:

We appreciate the thoughtful, thorough, and constructive reviews of the manuscript from reviewers and the associate editor. Both referees brought up several valid concerns and provide many excellent suggested edits and comments to address the problems. We have made major revisions to the manuscript in accordance with these suggestions and feel the paper has improved significantly.

Major changes to the manuscript include:

- A new title that better reflects the content and conclusions of the paper.
- Reformulation of the paper to a traditional format that includes a more comprehensive introduction, background, methods, results, and discussion section.
- Referee #1 noted that we focused our background and discussion too narrowly on New Zealand and as a result omitted a large body of literature on the subject of drainage network evolution in faulted landscapes and with respect to material strength heterogeneities in bedrock. We appreciate their literature suggestions and have incorporated these and other relevant works into the revised manuscript. We believe this provides a more holistic treatment of the subjects in question and helps to make the paper more universally relevant.
- Reanalysis of the faults and rivers orientations based on three geomorphic domains designated from study area topography rather than 8 arbitrary squares. We also include circular statistics in the figures and a new data table.
- The paper now includes 6 rather than the original 4 figures. We broke up figure 1 into a second figure with topographic swath profiles (Figure 2) and broke out the fault orientation data and the river orientation data into their own separate figures (Figures 3 and 4). We now include a chi map in figure 5.
- We reordered the discussion section such that the description of MFS landscape evolution comes last.
- We added a supplementary information file with a figure that includes a geologic map of the study area and the locations of low-temperature thermochronology samples from Collett et al (2019) as well as chi-elevation and distance-elevation plots of the Awatere and Clarence rivers.

The following table shows additional comments that we addressed. We present point by point referee comment and author response for the line edits, questions, and comments provided by the reviewers. We thank them for taking the time to provide these detailed suggestions and think that the revised manuscript is much improved as a result.

| Comment (Reviewer 1) | Response |
| --- | --- |
| I think the paper by Molnar et al 2007 cited here is more on the influence of rock weakening on erosion in general, rather than on any influence on river patterns along faults. | Reference to this paper was removed from the manuscript. |

| Since you cite Bishop 1995 here, who actually provides an in-depth examination of this issue, I would emphasise that the "can" is very important | We agree and have added a sentence that makes it clear that drainage anomalies, or unusual patterns in river planform, do not necessarily indicate recent river captures (lines 45 – 50). |
|---|---|
| "In the earliest phase of the Kaikoura orogeny": hard for outsiders to know when that is, perhaps it would be good to put xMa in brackets after this and elsewhere in the text. | Throughout the paper, we have added ages in brackets to show the specific timing of events mentioned in the text. |
| "There, the active faults are primarily strike-slip and have not generated the fault parallel, high-relief ranges (Fig.1) that would aide in the development of transverse drainage" - It can be readily observed in many mountain ranges, but also in field and roadcuts, or in the lab, or in numerical experiments, that transverse drainage develops easily, without needing the aide of faults. See Hovius 1996 for instance for a first review of this. | We agree with the reviewer and regret that in the original draft we wrote the word transverse here but we meant longitudinal. There were a few other instances of this unfortunate mistake in the original draft. We have corrected each of these instances in the revised manuscript. |
| **Comment (Reviewer 2)** | **Response** |
| Line 50: based on your later results, I suggest writing "the position and orientation of rivers" rather than only orientation. | We added these words to the sentence – line 55. |
| Lines 66-67: in this sentence you are listing all your analysis, so I do not think that saying "including" is appropriate here, as it gives the impression that there are more analysis than those on the list. I would rephrase to simply say "In this study, we present analysis on the topography, fluvial morphologies in planform and profile forms, and orientations of rivers compared to active and inactive faults" | We have rephrased according to the reviewer's suggestion – line 80 - 82. |
| At present, the last 3 paragraphs of the Geologic Setting read a bit convoluted because they go from making a general statement on the overall evolution, to talking about the present-day configuration and slip rates, to the early deformation phase, and the evolution from Late Miocene to today. I would suggest following a chronological order, so switching lines 88-94 to the end of the section. | We have revised section 2 of the paper (the Geologic Background) to include three separate sections: Geologic Setting, Plate Tectonic History and Study Area Topography. Hopefully these subheadings make the information presented less convoluted and more clear. Lines 85 – 155. |
| Line 90: how have these slip rate estimates been derived? GPS? Offset dated surfaces? A large number of studies are referred, but readers should not need to be familiar with those in order to have a general idea – a general statement saying "derived from…" would be helpful. | These slip rates were derived from offset dated features and this has now been added to the paper. Line 135. |
| Line 97: please be more specific with the geologic time you are referring to when saying "Early in the plate boundary history" (is it Late Oligocene, | We are now more specific with the geologic time period that we are referring to. |

| | |
|---|---|
| Early Miocene, Early to Mid Miocene…?). You could add a parenthesis specifying this before the coma. | |
| Line 98: what type of structures? Just saying "a few important structures" is vague. Figure 4 suggests that these were primarily thrusts and folds associated with them, but this information should be clearly presented in the geological setting, particularly given that it is going to be heavily included in the discussion. | We now more explicitly describe the structures in detail in the updated Section 2 (Geologic Background). |
| Line 104: again, I think the readers would benefit from greater clarity on the time you are referring to (25 Ma?). Also, to follow a clear chronological order, I would suggest that this sentence goes when the geological history is starting to be discussed, at the beginning of the 2nd paragraph. | We have added more details on the timing of events in the Geologic Background Section. |
| Line 108: here or when discussing current slip rates – could you provide with some estimates on the partitioning of vertical vs. lateral motion? "Lesser" is quite vague. | We have added details about the ratio of horizontal to vertical slip on the faults. Lines 126 – 128. |
| Line 110: readers would benefit from a brief statement describing how have the "estimates of timing, cumulative decrease in total offset, and increase in slip rates" have been derived, or at least what type of data set they come from. Also, could you please explain what is meant by "cumulative decrease in total offset"? I understand how an increase in offsets could inform about the time since fault activity started, but I am not sure how could a decrease in offsets inform of that, or how it could even be identified or resolved. | These details have been added and the sentences revised for clarity. Lines 135 – 143. |
| Given that the dividing line is arbitrary, and that many of the landscape features are transitional – is dividing the área in "domains" actually necessary? I suggest that the authors simply refer to the ENE and WSW parts of the study area, or include some other features (peaks, towns) as a point of reference, rather than making an arbitrary division that I also think complicates their interpretations in the following figures, given that this divide position does not actually correspond with any clear geomorphic boundaries. | We now more simply refer to the three geomorphic domains as eastern Marlborough north of the Hope fault, western Marlborough north of the Hope faul and south of the Hope fault. |
| The first two paragraphs of this section read a lot like information that should be on the Geological Setting section, given that, except the swath profiles, there is no "result" of analysis presented. I suggest moving these paragraphs that simply describe the landscape features from the DEM and the positions of the faults based on published | These paragraphs have been moved into their proper sections. Information about the DEM and fault database has been added to the text. |

| | |
|---|---|
| data, to a sub-section of the Geological Setting. Also, I see that the DEM used and the source of the faults map are listed in the caption of the figure, but this is important information that should be included on the main text, in a Methods section. | |
| Lines 154: can you use these slip rate estimates to infer minimum time since the Clarence started flowing to the SE in its lower reaches? Or using the offset to estimate the beginning of slip in the Kekerengu fault? This would better highlight the potential of drainage patterns on informing about tectonic evolution. | Yes, we now add this estimation to the text. Lines 292 – 296. |
| Line 166: the previous line mentions both the Awatere and the Clarence river, so it is not clear what river and what segment is referred to when saying "this segment", please be more specific. | We are now more specific about which river we refer to. |
| Lines 167-168: this short sentence says twice "in the headwaters of the Awatere river" | Reworded the sentences. |
| Figure 2a and 2b: these are two important figures for the paper's results, but it is often hard to follow the results because the figures are too small and cluttered, and two important features for the analysis, the relief and the faults, are displayed in other figures, making it harder to relate them to the drainage network. I suggest moving the faults and river orientation analysis (panels c and d) to another figure, and make this figure a bigger panel figure with 4 or 2 panels, | We have broken out the fault and river analyses into two separate figures (Figure 3 and Figure 4). |
| Lines 175-177: These sentences belong in the Geological Setting, they are not the results of this paper. | Moved to Geologic Background section. |
| Lines 180-183: This information is important, but belongs in a methods section. Also, please explicitly state whether you follow the same criteria as GNS to consider if a fault is active or inactive, and what do you mean by "mature" faults (ie. An inactive fault could be mature? For example if it was active for long enough to significantly weaken the bedrock). | Yes, we have now added a proper Methods Section. We explicitly state that we are following GNS criterion for fault activity. By "mature" we mean that the fault has had enough displacement/time to promote material strength weakening along the fault. We are more clear with this description and language in the revised draft. |
| Clearly A5 and A8 span both domains, so it is problematic to overlap the previous Inland Malborough vs. Kaikoura domains to this grid pattern. I understand the practicalities of diving the area in grids, but as they are right now, these grids are not truly representative of the different areas, and if anything, they could be masking some trends. | We have removed the arbitrary 8 squares and now perform the analysis in the 3 domains. |

| | |
|---|---|
| Line 186: why have these channel orders been selected? This needs a brief justification in the methods. | The new methods section has a better description and justification for the channel orders chosen and more properly describes the analysis methods. |
| Line 187: some description of what these "network segment and plotting routines" are and do is needed in the methods. | These have been added to the new methods section. |
| Line 186: again, this belongs in the methods, it is not a result. How have you done this normalization? | Yes, agreed. The description of our segment weighting process is now included in the new methods section. |
| I strongly suggest using a more quantitative, statistics-based way to assess the overlap of the inactive and active faults and river orientations | We have added circular statistics to this revised manuscript. |
| Lines 208-218: none of this are results from this study, this paragraph belongs in the introduction. | Moved to introduction. |
| Line 220: I suggest adding a lithological map of the study area, it would be very helpful for readers not familiar with this area of NZ but interested in your drainage evolution results. | We have added a geologic map showing different lithologies across the study site to a supplemental data file. This map also includes low-temperature thermochronology sample locations from Collett et al. (2019). |
| Lines 225-234: all this belongs in a Methods section. What "default values"? From what paper/software? | Yes, this information has been added to the new methods section. We include all useful information and no longer point to "default values". |
| Line 233: the chi-plots used to identify breaks in slope should be included in the supplementary information. | These are now included in the supplementary information. |
| Line 236: ksn should have units of m0.9 if a reference concavity of 0.45 (~0.5) is used. | Ksn has units of m as we used 0.5 as the reference concavity. |
| Lines 245-251 and 254-256: contextualizing this paper's findings with previous published studies belongs in the discussion, not in the results. | Moved to the discussion. |
| Line 292: and as they responded to the increase in uplift… | Added |
| Line 332-333: this should have been mentioned in the results. | It is now included in the results. |
| Line 364: I suggest adding "enough displacement to…or to produce significant relief" | We added this phrase. |
| Line 400: please do state explicitly what factors were investigated – many people read the conclusions of a paper before deciding whether to read it entirely or not, so this would be relevant information. | We have now updated the conclusions to state what was investigated. |
| Line 38: space missing between "e.g." and "Wobus" | Corrected |
| Line 17: for clarity, please insert "drainage" here, so that it reads "history of drainage capture and rearrangement" | We inserted the word drainage to this sentence. |
| Line 64: space missing between "e.g." and "King" | Corrected |

| | |
|---|---|
| Line 69: I suggest changing "complicated" for "complex", otherwise the word "complicated" is repeated 3 times in 6 lines. | Changed to complicated to avoid repeating the same word too many times. |
| Line 75: typo, "Puysegur" not "Puyseguer" | Corrected |
| Line 139: space missing between "e.g." and "Bishop" | Corrected |
| Line 409: it is "Philippe Steer" not "Phillipe Steere" | Corrected |